# The mechanism of variability in transcription start site selection

Libing Yu[1,2†], Jared T Winkelman[1,2,3†], Chirangini Pukhrambam[2,3], Terence R Strick[4,5,6], Bryce E Nickels[2,3*], Richard H Ebright[1,2*]

[1]Department of Chemistry, Rutgers University, Piscataway, United States; [2]Waksman Institute, Rutgers University, Piscataway, United States; [3]Department of Genetics, Rutgers University, Piscataway, United States; [4]Ecole Normale Supérieure, Institut de Biologie de l'Ecole Normale Supérieure (IBENS), CNRS, INSERM, PSL Research University, Paris, France; [5]Programme Equipe Labellisées, Ligue Contre le Cancer, Paris, France; [6]Institut Jacques Monod, CNRS, UMR7592, University Paris Diderot, Paris, France

**Abstract** During transcription initiation, RNA polymerase (RNAP) binds to promoter DNA, unwinds promoter DNA to form an RNAP-promoter open complex (RPo) containing a single-stranded 'transcription bubble,' and selects a transcription start site (TSS). TSS selection occurs at different positions within the promoter region, depending on promoter sequence and initiating-substrate concentration. Variability in TSS selection has been proposed to involve DNA 'scrunching' and 'anti-scrunching,' the hallmarks of which are: (i) forward and reverse movement of the RNAP leading edge, but not trailing edge, relative to DNA, and (ii) expansion and contraction of the transcription bubble. Here, using in vitro and in vivo protein-DNA photocrosslinking and single-molecule nanomanipulation, we show bacterial TSS selection exhibits both hallmarks of scrunching and anti-scrunching, and we define energetics of scrunching and anti-scrunching. The results establish the mechanism of TSS selection by bacterial RNAP and suggest a general mechanism for TSS selection by bacterial, archaeal, and eukaryotic RNAP.

DOI: https://doi.org/10.7554/eLife.32038.001

*For correspondence:
bnickels@waksman.rutgers.edu
(BEN);
ebright@waksman.rutgers.edu
(RHE)

†These authors contributed equally to this work

Competing interests: The authors declare that no competing interests exist.

## Introduction

During transcription initiation, RNA polymerase (RNAP) and one or more transcription initiation factor bind to promoter DNA through sequence-specific interactions with core promoter elements, unwind a turn of promoter DNA to form an RNAP-promoter open complex (RPo) containing an unwound 'transcription bubble,' and select a transcription start site (TSS). The distance between core promoter elements and the TSS can vary. TSS selection is a multi-factor process, in which the outcome reflects the contributions of promoter sequence and reaction conditions. TSS selection by bacterial RNAP and the bacterial transcription initiation factor σ involves four promoter-sequence determinants: (i) distance relative to the promoter −10 element (preference for TSS selection at the position 7 bp downstream of the promoter −10 element; *Aoyama and Takanami, 1985*; *Sørensen et al., 1993*; *Jeong and Kang, 1994*; *Liu and Turnbough, 1994*; *Walker and Osuna, 2002*; *Lewis and Adhya, 2004*; *Vvedenskaya et al., 2015*; *Winkelman et al., 2016a*; *Winkelman et al., 2016b*); (ii) identities of the template-strand nucleotide at the TSS and the template-strand nucleotide immediately upstream of the TSS (strong preference for a template-strand pyrimidine at the TSS and preference for a template-strand purine immediately upstream of the TSS; *Aoyama and Takanami, 1985*; *Sørensen et al., 1993*; *Jeong and Kang, 1994*; *Liu and Turnbough, 1994*; *Walker and Osuna, 2002*; *Lewis and Adhya, 2004*; *Vvedenskaya et al., 2015*; *Winkelman et al., 2016a*; *Winkelman et al., 2016b*); (iii) the promoter 'core recognition element,' a

**eLife digest** Genes store the information needed to build and repair cells. This information is written in a chemical code within the structure of DNA molecules. To make use of the information, cells copy sections of a gene into a DNA-like molecule called RNA. An enzyme called RNA polymerase makes RNA molecules from DNA templates in a process called transcription. RNA polymerase can only make RNA by attaching to DNA and separating the two strands of the DNA double helix. This creates a short region of single-stranded DNA known as a "transcription bubble".

RNA polymerase can start transcription at different distances from the sites where it initially attaches to DNA, depending on the DNA sequence and the cell's environment. It had not been known how RNA polymerase selects different transcription start sites in different cases. One hypothesis had been that differences in the size of the transcription bubble – the amount of unwound single-stranded DNA – could be responsible for differences in transcription start sites. For example, RNA polymerase could increase the size of the bubble through a process called "DNA scrunching", in which RNA polymerase pulls in and unwinds extra DNA from further along the gene.

Yu, Winkelman et al. looked for indicators of DNA scrunching to see whether it contributes to the selection of transcription start sites. By mapping the positions of the two edges of RNA polymerase relative to DNA, they saw that RNA polymerase pulls in extra DNA when selecting a transcription start site further from its initial attachment site. Next, by measuring the amount of DNA unwinding, they saw that RNA polymerase unwinds extra DNA when it selects a transcription start site further from its initial attachment site. This was the case for both RNA polymerase in a test tube and RNA polymerase in living bacterial cells. The results showed that DNA scrunching accounts for known patterns of selection of transcription start sites.

The findings hint at a common theory for the selection of transcription start sites across all life by DNA scrunching. Understanding these basic principles of biology reveals more about how cells work and how cells adapt to changing conditions. The experimental methods developed for mapping the positions of proteins on DNA and for measuring DNA unwinding will help scientists to learn more about other aspects of how DNA is stored, copied, read, and controlled.

DOI: https://doi.org/10.7554/eLife.32038.002

segment of nontemplate-strand sequence spanning the TSS that interacts sequence-specifically with RNAP (preference for nontemplate-strand G immediately downstream of the TSS; *Vvedenskaya et al., 2016*), and (iv) the promoter 'discriminator element,' a nontemplate-strand sequence immediately downstream of the promoter −10 element that interacts sequence-specifically with σ (preference for TSS selection at upstream positions for purine-rich discriminator sequences, and preference for TSS selection at downstream positions for pyrimidine-rich discriminator sequences; *Winkelman et al., 2016a, 2016b*). In addition to these four promoter-sequence determinants, the concentrations of initiating NTPs (*Sørensen et al., 1993*; *Liu and Turnbough, 1994*; *Walker and Osuna, 2002*; *Vvedenskaya et al., 2015*; *Wilson et al., 1992*; *Qi and Turnbough, 1995*; *Tu and Turnbough, 1997*; *Walker et al., 2004*; *Turnbough, 2008*; *Turnbough and Switzer, 2008*) and DNA topology (*Vvedenskaya et al., 2015*) also influence TSS selection.

It has been hypothesized that variability in the distance between core promoter elements and the TSS is accommodated by DNA 'scrunching' and 'anti-scrunching,' the defining hallmarks of which are: (i) forward and reverse movements of the RNAP leading edge, but not the RNAP trailing edge, relative to DNA, and (ii) expansion and contraction of the transcription bubble (*Vvedenskaya et al., 2015*; *Winkelman et al., 2016a, 2016b*; *Vvedenskaya et al., 2016*; *Robb et al., 2013*). In previous work, we showed that TSS selection exhibits the first hallmark of scrunching in vitro (*Winkelman et al., 2016a*). Here, we show that TSS selection also exhibits the first hallmark of scrunching in vivo, show that TSS selection exhibits the second hallmark of scrunching and anti-scrunching, and define the energetics of scrunching and anti-scrunching.

## Results and discussion

### TSS selection exhibits first hallmark of scrunching–movements of RNAP leading edge but not RNAP trailing edge–both in vitro and in vivo

In our prior work, we demonstrated that bacterial TSS selection in vitro exhibits the first hallmark of scrunching by defining, simultaneously, the TSS, the RNAP leading-edge position, and RNAP trailing-edge position for transcription complexes formed on a library of $10^6$ promoter sequences (*Winkelman et al., 2016a*). We used RNA-seq to define the TSS, and we used unnatural-amino-acid-mutagenesis, incorporating the photoactivatable amino acid *p*-benzoyl-L-phenylalanine (Bpa), and protein-DNA photocrosslinking to define RNAP-leading-edge and trailing-edge positions (*Winkelman et al., 2016a*). The results showed that the discriminator element (*Haugen et al., 2006*; *Feklistov et al., 2006*) influences TSS selection and does so through effects on sequence-specific σ-DNA interaction that select between two alternative paths of the DNA nontemplate strand (*Winkelman et al., 2016a*). The results further showed that, as the TSS changes for different discriminator sequences, the RNAP-leading-edge position changes, but the RNAP-trailing-edge position does not change (*Winkelman et al., 2016a*). For example, replacing a GGG discriminator by a CCT discriminator causes a 2 bp downstream change in TSS (from the position 7 bp downstream of the −10 element to the position 9 bp downstream of the −10 element, due to differences in sequence-specific σ-DNA interaction that result in different paths of the DNA nontemplate strand), causes a 2 bp downstream change in RNAP leading-edge position, but does not cause a change in RNAP trailing-edge position (*Figure 1A*).

Here, to determine whether bacterial TSS selection in vivo also exhibits the first hallmark of scrunching, we adapted the above unnatural-amino-acid-mutagenesis and protein-DNA-photocrosslinking procedures to define RNAP leading-edge and trailing-edge positions in TSS selection in living cells (*Figure 1*, *Figure 1—figure supplements 1–2*). We developed approaches to assemble, trap, and UV-irradiate RPo formed by a Bpa-labeled RNAP derivative in living cells, to extract crosslinked material from cells, and and to map crosslinks at single-nucleotide resolution (*Figure 1—figure supplement 1*). In order to assemble, trap, and UV-irradiate RPo in living cells, despite the presence of high concentrations of initiating substrates that rapidly convert RPo into transcribing complexes, we used a mutationally inactivated RNAP derivative, β'D460A, that lacks a residue required for binding of the RNAP-active-center catalytic metal ion and initiating substrates (*Zaychikov et al., 1996*) (*Figure 1—figure supplements 1–2*). Control experiments confirm that, in vitro, in both the absence and presence of initiating substrates, the mutationally inactivated RNAP derivative remains trapped in RPo, exhibiting the same pattern of leading-edge and trailing-edge crosslinks as for wild-type RNAP in the absence of initiating substrates (*Figure 1—figure supplement 2*). In order to introduce Bpa at the leading-edge and trailing-edge of RPo in living cells, we co-produced, in *Escherichia coli*, a Bpa-labeled, decahistidine-tagged, mutationally inactivated RNAP derivative in the presence of unlabeled, untagged, wild-type RNAP, using a three-plasmid system comprising (i) a plasmid carrying a gene for RNAP β' subunit that contained a nonsense codon at the site for incorporation of Bpa, the β'D460A mutation, and a decahistidine coding sequence; (ii) a plasmid carrying genes for an engineered Bpa-specific nonsense-suppressor tRNA and an engineered Bpa-specific aminoacyl-tRNA synthase (*Chin et al., 2002*); and (iii) a plasmid containing a promoter of interest (*Figure 1—figure supplement 1A*). (Using this merodiploid system, with both a plasmid-borne mutant gene for β' subunit and a chromosomal wild-type gene for β' subunit, enabled analysis of the mutationally inactivated RNAP derivative without loss of viability.) In order to perform RNAP-DNA crosslinking and to map resulting RNAP-DNA crosslinks, we then grew cells in medium containing Bpa, UV-irradiated cells, lysed cells, purified crosslinked material using immobilized metal-ion-affinity chromatography targeting the decahistidine tag on the Bpa-labeled, decahistidine-tagged, mutationally inactivated RNAP derivative, and mapped crosslinks using primer extension (*Figure 1—figure supplement 1B*). The results showed an exact correspondence of crosslinking patterns in vitro and in vivo (*Figure 1B*, 'in vitro' vs. 'in vivo' lanes). The RNAP leading edge crosslinked 2 bp further downstream on CCT than on GGG, whereas the RNAP trailing edge crosslinked at the same positions on CCT and GGG (*Figure 1B*). We conclude that TSS selection in vivo shows the first hallmark of scrunching.

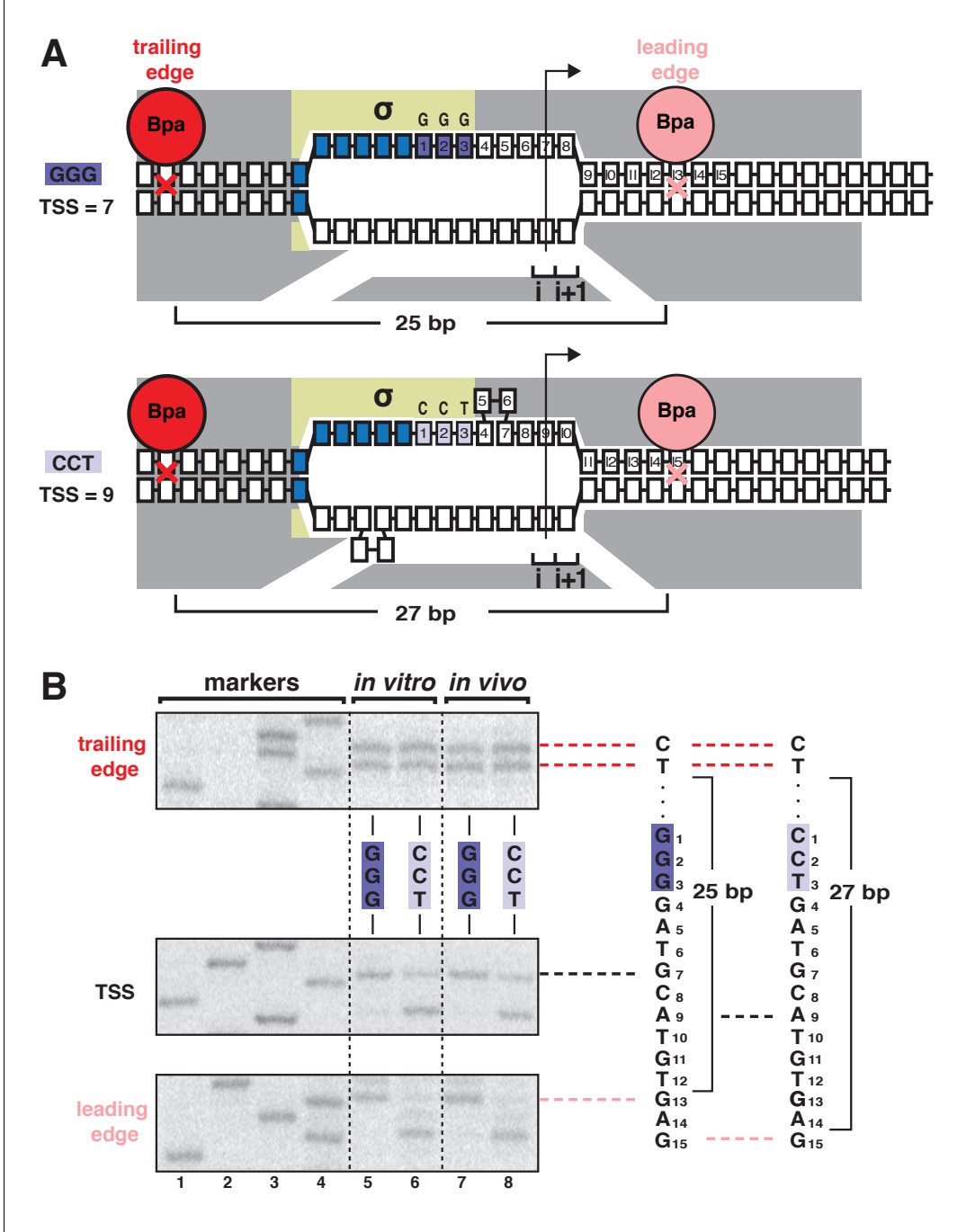

**Figure 1.** TSS selection exhibits first hallmark of scrunching–movement of RNAP leading edge but not RNAP trailing edge–both in vitro and in vivo. (**A**) RNAP leading-edge and trailing-edge positions at promoters having GGG discriminator (TSS 7 bp downstream of −10 element; top) and CCT discriminator (TSS 9 bp downstream of −10 element; bottom). Changes in TSS selection result from changes in discriminator-sequence-dependent DNA scrunching. Gray, RNAP; yellow, σ; blue, −10-element nucleotides; dark purple, GGG-discriminator nucleotides; light purple, CCT-discriminator nucleotides; i and i + 1, NTP binding sites; arrow, TSS; boxes, DNA nucleotides (nontemplate-strand nucleotides above template-strand nucleotides; nucleotides downstream of −10 element numbered); red, trailing-edge Bpa and nucleotide crosslinked to Bpa; pink, leading-edge Bpa and nucleotide crosslinked to Bpa. Scrunching is indicated by bulged-out nucleotides. Distance between leading-edge and trailing-edge crosslinks is indicated below RNAP. (**B**) RNAP trailing-edge crosslinking (top), TSS (middle), and RNAP leading-edge crosslinking (bottom) for promoters having GGG discriminator and CCT discriminator, in vitro (lanes 5–6) and in vivo (lanes 7–8). Horizontal dashed lines relate bands on gel (left) to nucleotide sequences (right).
DOI: https://doi.org/10.7554/eLife.32038.003

The following figure supplements are available for figure 1:

*Figure 1 continued on next page*

*Figure 1 continued*

**Figure supplement 1.** Unnatural-amino-acid mutagenesis and protein-DNA photocrosslinking in vivo.
DOI: https://doi.org/10.7554/eLife.32038.004
**Figure supplement 2.** Mutationally inactivated RNAP derivative traps RPo in presence of NTPs, enabling protein-DNA photocrosslinking of RPo in presence of NTPs.
DOI: https://doi.org/10.7554/eLife.32038.005

## TSS selection exhibits second hallmark of scrunching–changes in size of transcription bubble

The results in *Figure 1* establish that TSS selection in vitro and in vivo exhibits the first hallmark of scrunching. However, definitive demonstration that TSS selection involves scrunching also requires demonstration of the second hallmark of scrunching: that is, changes in transcription-bubble size. To determine whether bacterial TSS selection exhibits the second hallmark of scrunching we used a magnetic-tweezers single-molecule DNA-nanomanipulation assay that enables detection of RNAP-dependent DNA unwinding with near-single-base-pair spatial resolution and sub-second temporal resolution (*Revyakin et al., 2004*, *2005*, *2006*) to assess whether TSS selection correlates with transcription-bubble size for GGG and CCT promoters (*Figure 2*). The results indicate that transition amplitudes for RNAP-dependent DNA unwinding upon formation of RPo with CCT are larger than those for formation of RPo with GGG, on both positively and negatively supercoiled templates (*Figure 2B*, left and center). Transition-amplitude histograms confirm that transition amplitudes with CCT are larger than with GGG, on both positively and negatively supercoiled templates (*Figure 2B*, right). By combining the results with positively and negatively supercoiled templates to deconvolve effects of RNAP-dependent DNA unwinding and RNAP-dependent compaction (*Revyakin et al., 2004*, *2005*, *2006*), we find a 2 bp difference in RNAP-dependent DNA unwinding for CCT vs. GGG (*Figure 2C*), corresponding exactly to the 2 bp difference in TSS selection (*Figure 1B*). We conclude that TSS selection shows the second hallmark of scrunching.

## TSS selection downstream and upstream of modal TSS involves scrunching and anti-scrunching, respectively: forward and reverse movements of RNAP leading edge

According to the hypothesis that TSS selection involves scrunching or anti-scrunching, TSS selection at the most frequently observed, modal TSS position (7 bp downstream of −10 element for majority of discriminator sequences, including GGG) involves neither scrunching nor anti-scrunching, TSS selection downstream of the modal position involves scrunching (transcription-bubble expansion), and TSS selection upstream of the modal position involves anti-scrunching (*Vvedenskaya et al., 2015*; *Winkelman et al., 2016a*, *2016b*; *Vvedenskaya et al., 2016*; *Robb et al., 2013*). The results in *Figures 1–2* apply to the modal TSS position and a TSS position 2 bp downstream of the modal TSS position. To generalize and extend the results to a range of different TSS positions, including a position upstream of the modal TSS position expected to involve anti-scrunching, we exploited the ability of oligoribonucleotide primers ('nanoRNAs'; *Goldman et al., 2011*) to program TSS selection (*Figure 3A*, *Figure 3—figure supplement 1*). We analyzed a consensus bacterial promoter, *lac*-CONS, and used four ribotrinucleotide primers, UGG, GGA, GAA, and AAU, to program TSS selection at positions 6, 7, 8, and 9 bp downstream of the −10 element (*Figure 3A*). Experiments analogous to those in *Figure 1* show a one-for-one, bp-for-bp correlation between primer-programmed changes in TSS and changes in RNAP-leading-edge position. The leading-edge crosslink positions with the four primers differed in single-nucleotide increments, but the trailing-edge crosslink positions were the same (*Figure 3B*). With the primer GGA, which programs TSS selection at the modal position (7 bp downstream of −10 element for this discriminator sequence), the leading-edge crosslinks were exactly as in experiments with no primer (*Figure 3—figure supplement 1*). With primers GAA and AAU, which program TSS selection 1 and 2 bp downstream (positions 8 and 9), leading-edge crosslinks were 1 and 2 bp downstream of crosslinks with GGA (*Figure 3B*). With primer UGG, which programs TSS selection 1 bp upstream (position 6), leading-edge crosslinks were 1 bp upstream of crosslinks with GGA (*Figure 3B*). The results show that successive single-base-pair changes in TSS selection are matched by successive single-base-pair changes in RNAP leading-edge

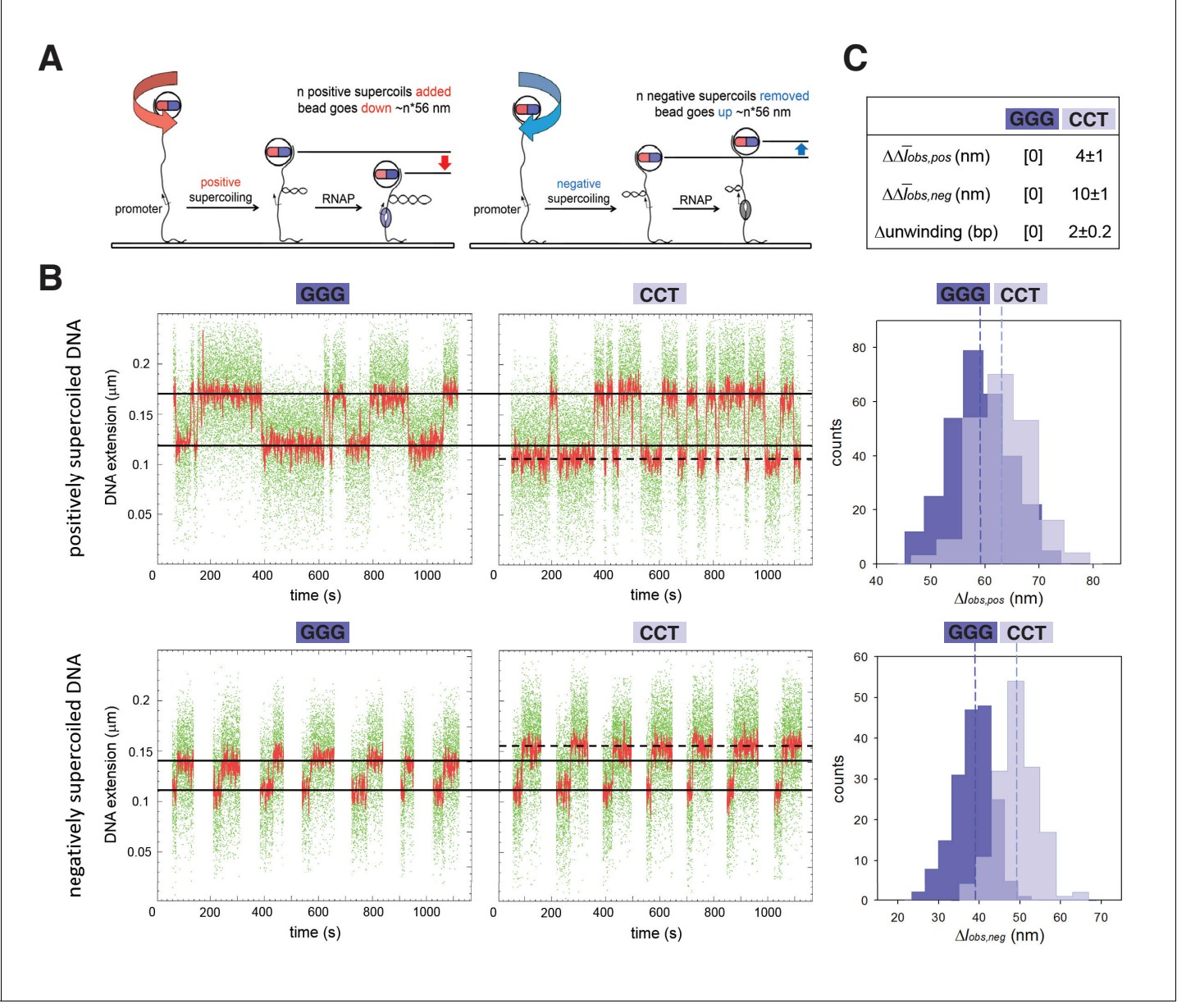

**Figure 2.** TSS selection exhibits second hallmark of scrunching–change in size of transcription bubble. (**A**) Magnetic-tweezers single-molecule DNA nanomanipulation (*Revyakin et al., 2004*; *Revyakin et al., 2005*; *Revyakin et al., 2006*). End-to-end extension (l) of a mechanically stretched, positively supercoiled (left), or negatively supercoiled (right), DNA molecule is monitored. Unwinding of $n$ turns of DNA by RNAP results in compensatory gain of $n$ positive supercoils or loss of $n$ negative supercoils, and movement of the bead by $n*56$ nm. (**B**) Single-molecule time traces and transition-amplitude histograms for RPo at promoters having GGG discriminator or CCT discriminator. Upper subpanel, positively supercoiled DNA; lower subpanel, negatively supercoiled DNA. Green points, raw data (30 frames/s); red points, averaged data (1 s window); horizontal black lines, wound and unwound states of GGG promoter; dashed horizontal black line, unwound state of CCT promoter; vertical dashed lines, means; $\Delta l_{obs,pos}$, transition amplitude with positively supercoiled DNA; $\Delta l_{obs,neg}$, transition amplitude with negatively supercoiled DNA; $\bar{\Delta l}_{obs,pos}$, mean $\Delta l_{obs,pos}$; $\bar{\Delta l}_{obs,neg}$, mean $\Delta l_{obs,neg}$. (**C**) Differences in $\bar{\Delta l}_{obs,pos}$, $\bar{\Delta l}_{obs,neg}$, and DNA unwinding between GGG-discriminator promoter and CCT-discriminator promoter (means ± SEM).

DOI: https://doi.org/10.7554/eLife.32038.006

position. We conclude that the first hallmark of scrunching is observed for a full range of TSS positions, including, importantly, a position upstream of the modal TSS expected to involve anti-scrunching.

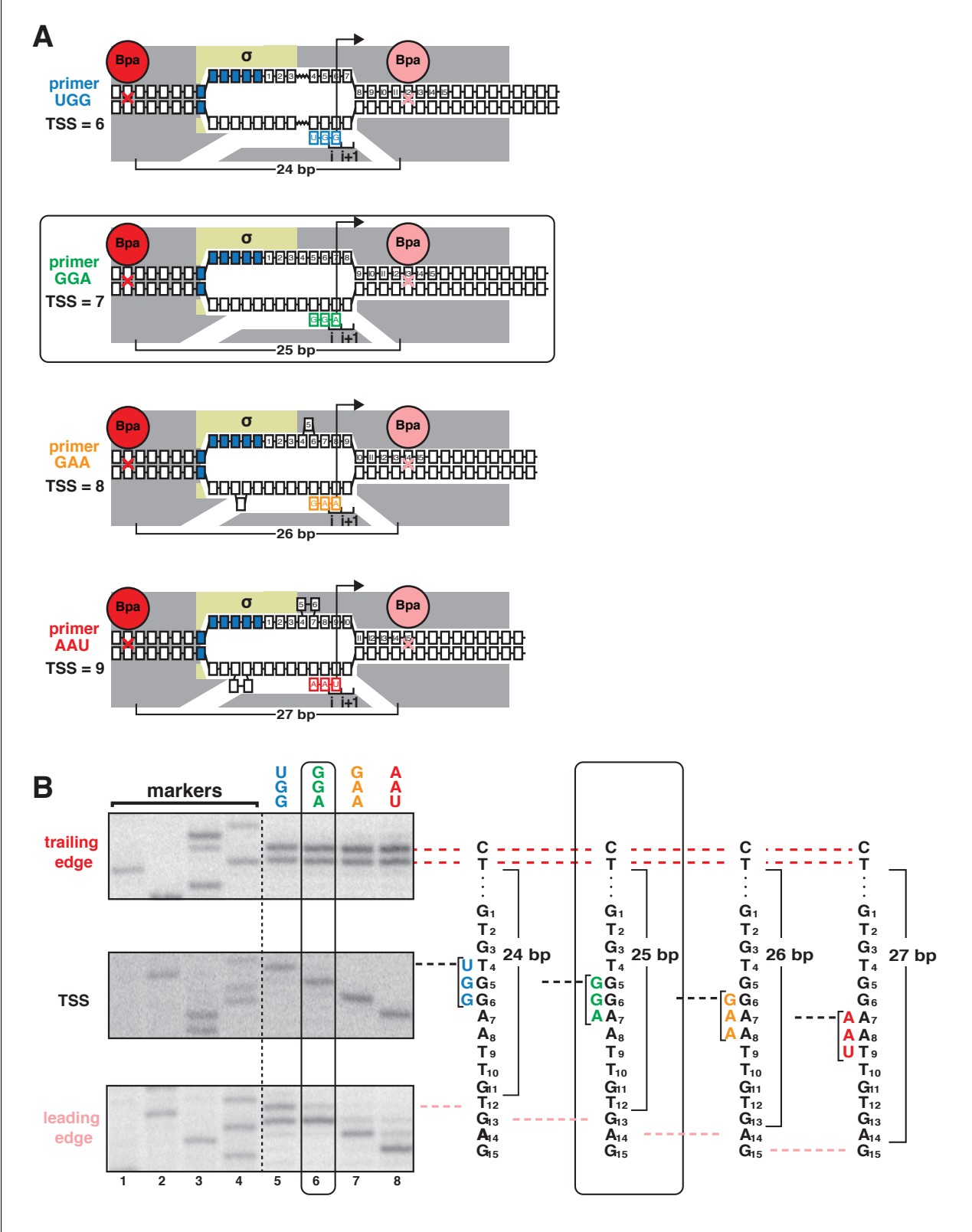

**Figure 3.** TSS selection downstream and upstream of the modal TSS involves, respectively, forward and reverse movements of RNAP leading edge. (A) Ribotrinucleotide primers program TSS selection at positions 6, 7, 8, and 9 bp downstream of −10 element (UGG, GGA, GAA, and AAU). Cyan, green, orange, and red denote primers UGG, GGA, GAA, and AAU, respectively. Rectangle with rounded corners highlights case of primer GGA, which programs TSS selection at same position as in absence of primer (7 bp downstream of −10 element). Other colors as in **Figure 1A**. (B) Use of protein-

*Figure 3 continued on next page*

*Figure 3 continued*

DNA photocrosslinking to define RNAP leading-edge and trailing-edge positions in vitro. RNAP trailing-edge crosslinking (top), TSS (middle), and RNAP leading-edge crosslinking (bottom) with primers UGG, GGA, GAA, and AAU (lanes 5–8). Horizontal dashed lines relate bands on gel (left) to nucleotide sequences (right).

DOI: https://doi.org/10.7554/eLife.32038.007

The following figure supplement is available for figure 3:

**Figure supplement 1.** Protein-DNA photocrosslinking in primer-programmed TSS selection: primer GGA yields same pattern of RNAP leading-edge and trailing-edge crosslinking as in absence of primer.

DOI: https://doi.org/10.7554/eLife.32038.008

## TSS selection downstream and upstream of modal TSS involves scrunching and anti-scrunching, respectively: increases and decreases in RNAP-dependent DNA unwinding

We next used magnetic-tweezers single-molecule DNA-nanomanipulation to analyze primer-programmed TSS selection. To enable single-base-pair resolution, we reduced the DNA-tether length from 2.0 kb to 1.3 kb, thereby reducing noise due to compliance (*Figure 4—figure supplement 1*; see *Revyakin et al., 2005*). The resulting transition amplitudes, transition-amplitude histograms, and RNAP-dependent DNA unwinding values for TSS selection with saturating concentrations of the four primers show a one-for-one, base-pair-for-base-pair correlation between primer-programmed changes in TSS and changes in RNAP-dependent DNA unwinding (*Figure 4*). With primer GGA, which programs TSS selection at the modal position (7 bp downstream of the −10 element for this discriminator sequence), DNA unwinding was exactly as in experiments with no primer (*Figure 4—figure supplement 2*). With primers GAA and AAU, which program TSS selection 1 and 2 bp further downstream (positions 8 and 9), DNA unwinding was ~1 and ~2 bp greater than with GGA (*Figure 4*). With primer UGG, which programs TSS selection 1 bp upstream, DNA unwinding was ~1 bp less than that in experiments with GGA (*Figure 4*). The results show that successive single-base-pair changes in TSS selection are matched by successive single-base-pair changes in DNA unwinding for a full range of TSS positions including, importantly, a position upstream of the modal TSS expected to involve anti-scrunching. Taken together, the results of protein-DNA photocrosslinking (*Figure 3*, *Figure 3—figure supplement 1*) and DNA-nanomanipulation (*Figure 4*, *Figure 4—figure supplement 2*) demonstrate, definitively, the scrunching/anti-scrunching hypothesis for TSS selection.

### Energetic costs of scrunching and anti-scrunching

To quantify the energetic costs of scrunching and anti-scrunching, we measured primer-concentration dependences of lifetimes of unwound states (*Figures 5–6*, *Figure 6—figure supplement 1*). For each primer, increasing the primer concentration increases the lifetime of the unwound state ($t_{unwound}$), as expected for coupled equilibria of promoter unwinding, promoter scrunching, and primer binding (*Figures 5–6*). The results in *Figure 6D* show that the slopes of plots of mean $t_{unwound}$ ($\bar{t}_{unwound}$) vs. primer concentration differ for different primers. Fitting the results to the equation describing the coupled equilibria (*Figure 6C*) yields values of $K_{NpNpN}$, $\Delta G_{NpNpN}$, $K_{scrunch}$, and $\Delta G_{scrunch}$ for the four primers (*Figure 6E*, *Figure 6—figure supplement 1*). The results indicate that scrunching by 1 bp requires 0.7 kcal/mol, scrunching by 2 bp requires 1.7 kcal/mol, and anti-scrunching by 1 bp requires 1.8 kcal/mol (*Figure 6E*, *Figure 6—figure supplement 1*).

The results provide the first experimental determination of the energetic costs of scrunching and anti-scrunching in any context. We hypothesize that energetic costs on the same scale, ~0.7–1.8 kcal/mol per scrunched bp, also apply in the structurally and mechanistically related scrunching that occurs during initial transcription by RNAP (*Revyakin et al., 2006*; *Kapanidis et al., 2006*). We note that, according to this hypothesis, the scrunching by ~10 bp that occurs during initial transcription (*Revyakin et al., 2006*) results in an increase in the state energy of the transcription initiation complex by a total of ~7–18 kcal/mol (~10 x ~0.7–1.8 kcal/mol). This is an increase in state energy potentially sufficient to yield a 'stressed intermediate' (*Revyakin et al., 2006*; *Straney and Crothers, 1987*) having scrunching-dependent 'stress' comparable to the free energies of RNAP-promoter and RNAP-initiation-factor interactions that anchor RNAP at a promoter (~7–9 kcal/mol for sequence-specific component of RNAP-promoter interaction and ~13 kcal/mol for RNAP-initiation-factor

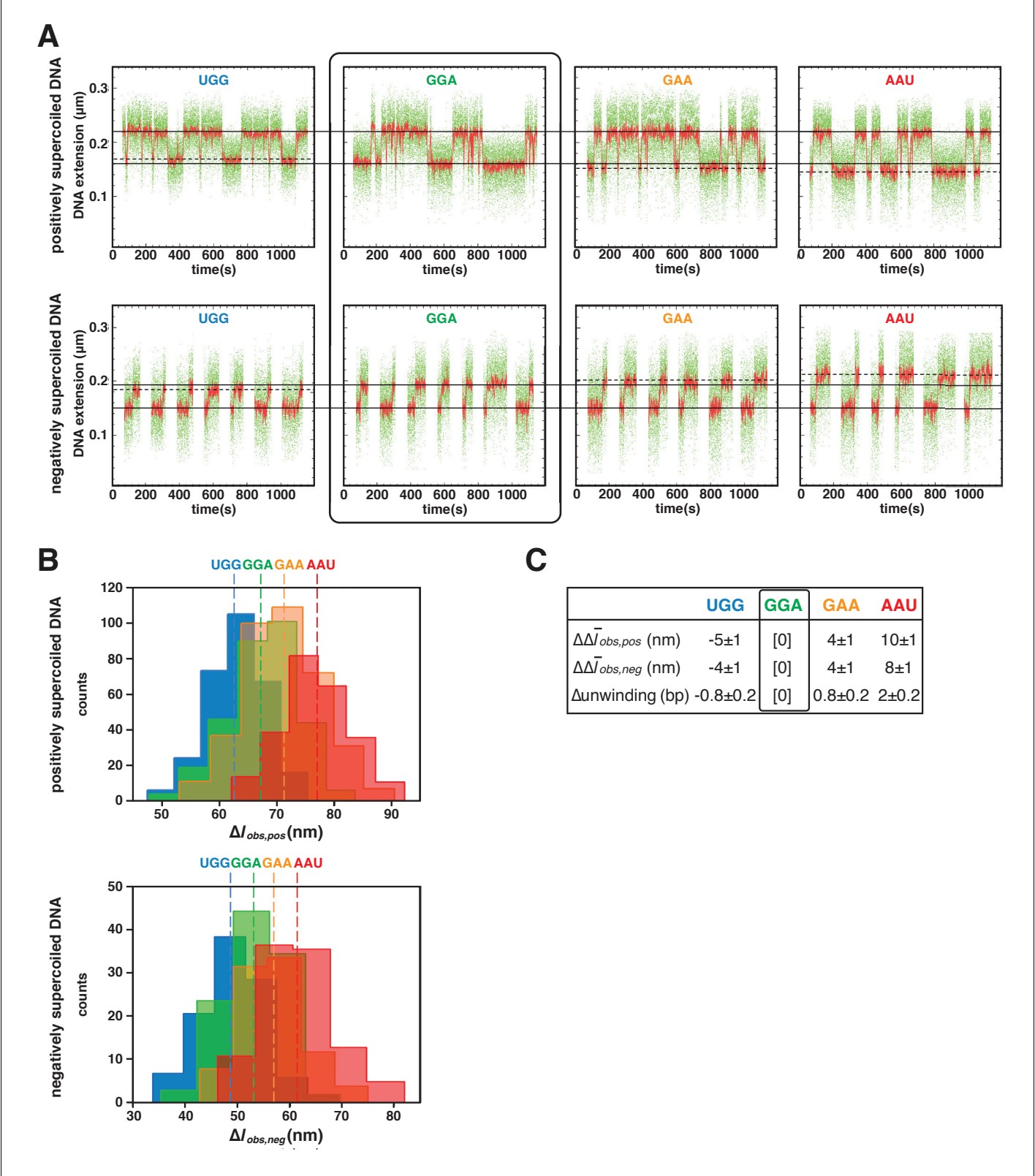

**Figure 4.** TSS selection downstream and upstream of the modal TSS involves, respectively: increases and decreases in RNAP-dependent DNA unwinding. (**A**) Use of single-molecule DNA nanomanipulation to define RNAP-dependent DNA unwinding. Single-molecule time traces with primers UGG, GGA, GAA, and AAU (positively supercoiled DNA in upper panel; negatively supercoiled DNA in lower panel). Rectangle with rounded corners highlights case of primer GGA, which programs TSS selection at position 7. Colors as in *Figure 2B*. (**B**) Transition-amplitude histograms (positively

*Figure 4 continued on next page*

*Figure 4 continued*

supercoiled DNA in upper panel; negatively supercoiled DNA in lower panel). (C) Differences in $\Delta \bar{l}_{obs,pos}$, $\Delta \bar{l}_{obs,neg}$, and DNA unwinding (bottom panel) with primers UGG, GGA, GAA, and AAU (means ± SEM).

DOI: https://doi.org/10.7554/eLife.32038.009

The following figure supplements are available for figure 4:

**Figure supplement 1.** Single-molecule DNA-nanomanipulation: shorter DNA fragment enables detection of RNAP-dependent DNA unwinding with single-base-pair resolution.

DOI: https://doi.org/10.7554/eLife.32038.010

**Figure supplement 2.** Single-molecule DNA-nanomanipulation: analysis of primer-programmed TSS selection with primer GGA.

DOI: https://doi.org/10.7554/eLife.32038.011

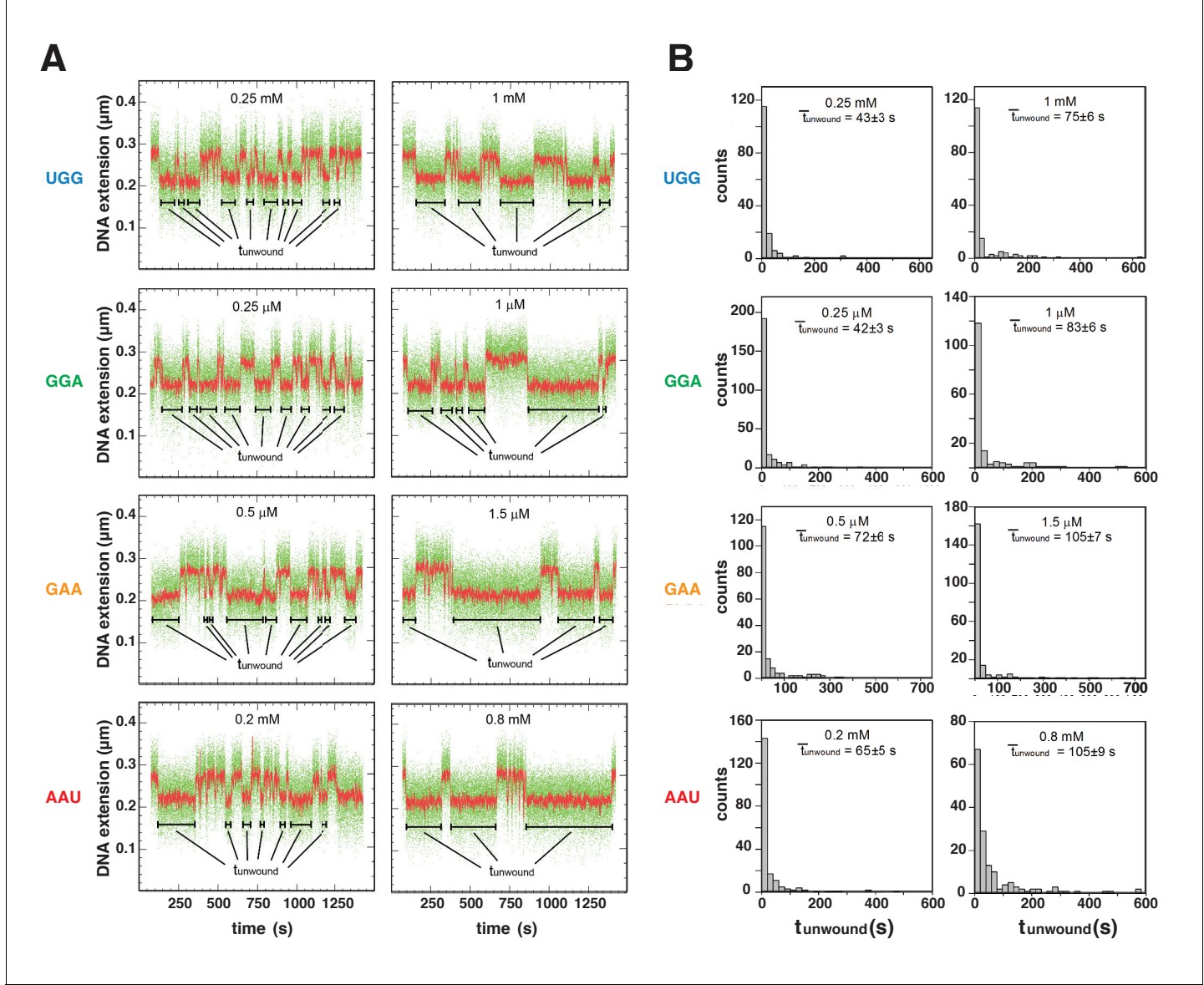

**Figure 5.** Energetic costs of scrunching and anti-scrunching: primer-concentration dependences of unwound-state lifetimes with primers UGG, GGA, GAA, and AAU. (A) Single-molecule time traces at low (left) and high (right) primer concentrations. Black bars, lifetimes of unwound states ($t_{unwound}$). Colors as in *Figure 2B*. (B) $t_{unwound}$ distributions and mean $t_{unwound}$ ($\bar{t}_{unwound}$). $\bar{t}_{unwound}$ increases with increasing primer concentrations.

DOI: https://doi.org/10.7554/eLife.32038.012

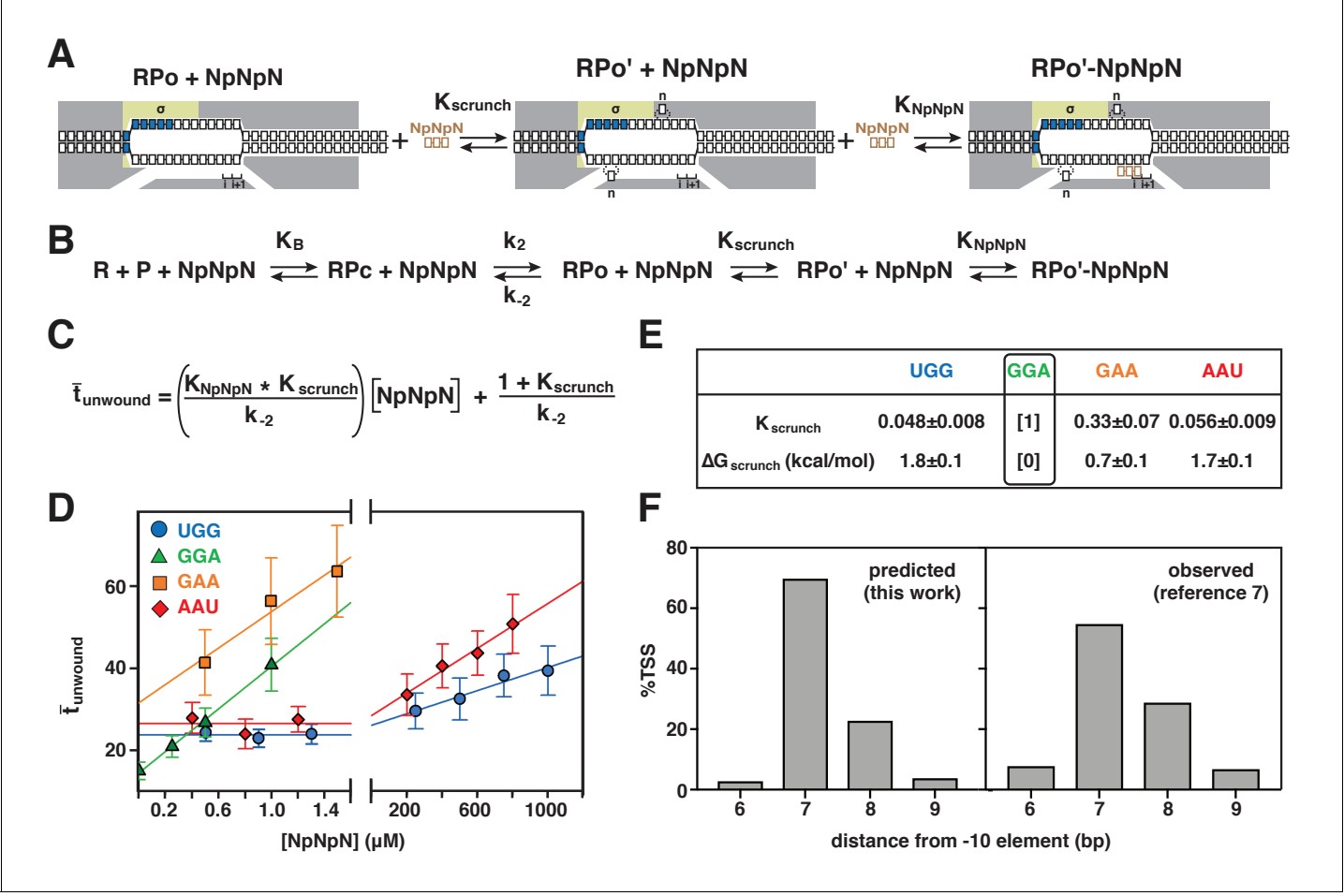

**Figure 6.** Energetic costs of scrunching and anti-scrunching: calculation of energetic costs and relationship between energetic costs and range and relative utilization of TSS positions. (**A**) Coupled equilibria for promoter scrunching or anti-scrunching ($K_{scrunch}$) and primer binding ($K_{NpNpN}$). (**B**) Equations for promoter binding ($K_B$), promoter unwinding ($k_2/k_{-2}$), promoter scrunching or anti-scrunching ($K_{scrunch}$), and primer binding ($K_{NpNpN}$). (**C**) Relationship between $\bar{t}_{unwound}$, $K_{scrunch}$, $K_{NpNpN}$, and primer concentration. (**D**) Dependences of mean lifetimes of unwound states ($\bar{t}_{unwound}$) on primer concentration for primers UGG, GGA, GAA, and AAU (means ± SEM). (**E**) Values of $K_{scrunch}$ and $\Delta G_{scrunch}$ calculated by fitting data in (**D**) to equation in (**C**) using $k_{-2} = 1/\bar{t}_{unwound}$ in absence of primer (*Revyakin et al., 2004*) (means ± SEM). Colors as in *Figure 3*. (**F**) TSS distributions predicted by Boltzmann-distribution probabilities for $\Delta G_{scrunch}$ values in (**E**) (left) and observed in analysis of comprehensive library of TSS-region sequences in *Vvedenskaya et al., 2015* (right).

DOI: https://doi.org/10.7554/eLife.32038.013

The following figure supplement is available for figure 6:

**Figure supplement 1.** Single-molecule DNA-nanomanipulation: calculation of $K_{scrunch}$ and $\Delta G_{scrunch}$ in primer-programmed TSS selection with primers UGG, GGA, GAA, and AAU.

DOI: https://doi.org/10.7554/eLife.32038.014

interaction; 24) and therefore is an increase in state energy potentially sufficient to pay energetic costs of breaking RNAP-promoter and RNAP-initiation-factor interactions in the transition from transcription initiation to transcription elongation.

## Energetic costs of scrunching and anti-scrunching in TSS selection explain range and relative utilization of TSS positions

The $\Delta G_{scrunch}$ values for bacterial TSS selection obtained in this work account for the range of TSS positions and the relative utilization of different TSS positions in bacterial transcription initiation. The $\Delta G_{scrunch}$ values for TSS selection at positions 6, 7, 8, and 9 of a promoter where the modal TSS is

position 7 (0–1.8 kcal/mol) all are less than or comparable to $3k_BT$ (~2 kcal/mol), where $k_B$ is the Bolztmann constant and T is temperature in °K, indicating that TSS selection at these positions requires no energy beyond energy available in the thermal bath. Indeed, the probabilities of TSS selection at positions 6, 7, 8, and 9 as observed in a comprehensive analysis of TSS-region sequences (8%, 55%, 29%, 7%; *Vvedenskaya et al., 2015*) can be predicted from the Boltzmann-distribution probabilities for the $\Delta G_{scrunch}$ values for TSS selection at these positions (3%, 70%, 23%, 4%; *Figure 6F*). The finding that values of $\Delta G_{scrunch}$ for scrunching and anti-scrunching in TSS selection are ~1 kcal mol$^{-1}$ bp$^{-1}$ and ~2 kcal mol$^{-1}$ bp$^{-1}$, respectively, implies that TSS selection at positions >2 bp downstream or >1 bp upstream of the modal position would exceed the energy fluctuations available to 99% of molecules at 20–37°C, and therefore explains the observation that TSS selection >2 bp downstream or >1 bp upstream of the modal position occurs rarely (*Vvedenskaya et al., 2015*).

## Unified mechanism of TSS selection by multisubunit RNAP

TSS selection by archaeal RNAP, eukaryotic RNAP I, eukaryotic RNAP II from most species, and eukaryotic RNAP III involves the same range of TSS positions as TSS selection by bacterial RNAP (positions ± 2 bp from the modal TSS; *Learned and Tjian, 1982*; *Samuels et al., 1984*; *Thomm and Wich, 1988*; *Reiter et al., 1990*; *Fruscoloni et al., 1995*; *Zecherle et al., 1996*). We propose that TSS selection by all of these enzymes is mediated by scrunching and anti-scrunching driven by energy available in the thermal bath. In contrast, TSS selection by *S. cerevisiae* RNAP II involves a range of TSS positions of 10 s to 100 s of bp (long-range TSS scanning; *Giardina and Lis, 1993*; *Kuehner and Brow, 2006*). We propose that TSS scanning by *S. cerevisiae* RNAP II also is mediated by scrunching and anti-scrunching, but, in this case, involves not only energy from the thermal bath, but also energy from the ATPase activity of RNAP II transcription factor TFIIH (*Sainsbury et al., 2015*). This proposal could account for the ATP-dependent, TFIIH-dependent cycles of DNA compaction and de-compaction of 10 s to 100 s of bp observed in single-molecule optical-tweezer analyses of TSS scanning by *S. cerevisiae* RNAP II (*Fazal et al., 2015*).

# Materials and methods

## Key resources table

| Reagent type (species) or resource | Designation | Source or reference | Identifiers | Additional information |
|---|---|---|---|---|
| strain, strain background (*E. coli*) | BL21(DE3) | New England BioLabs | Cat# C2527I | |
| strain, strain background (*E. coli*) | NiCo21(DE3) | New England BioLabs | Cat# C2529H | |
| strain, strain background (*T. thermophilus*) | HB8 | ATCC | Cat# ATCC 27634 | |
| recombinant DNA reagent | pUC18 | Thermo-Fisher | | |
| recombinant DNA reagent | pUC18-T20C2-lacCONS-CCT (plasmid) | this paper | | progenitors: PCR of *T. thermophilus* HB8, pUC18 |
| recombinant DNA reagent | pUC18-T20C2-lacCONS-CCT (plasmid) | this paper | | progenitors: PCR of *T. thermophilus* HB8, pUC18 |
| recombinant DNA reagent | pUC18-T20C2-lacCONS-CCT (plasmid) | this paper | | progenitors: PCR of *T. thermophilus* HB8, pUC18 |
| recombinant DNA reagent | pVS10 (plasmid) | PMID: 12511572 | | |
| recombinant DNA reagent | pIA900 (plasmid) | PMID: 25665556 | | |
| recombinant DNA reagent | pIA900-beta'R1148Bpa (plasmid) | PMID: 26257284 | | |
| recombinant DNA reagent | pIA900-beta'T48Bp | PMID: 26257284 | | |
| recombinant DNA reagent | pIA900-beta'R1148Bpa;beta'D460A | this paper | | progenitor: pIA900 |
| recombinant DNA reagent | pIA900-beta'T48Bpa;beta'D460A | this paper | | progenitor: pIA900 |
| recombinant DNA reagent | pEVOL-pBpF (plasmid) | PMID: 12154230; Addgene | cat# 31190 | |

*Continued on next page*

Continued

| Reagent type (species) or resource | Designation | Source or reference | Identifiers | Additional information |
|---|---|---|---|---|
| sequence-based reagent | UGG (oligoribonucleotide) | Trilink Biotechnologies | | |
| sequence-based reagent | GGA (oligoribonucleotide) | Trilink Biotechnologies | | |
| sequence-based reagent | GAA (oligoribonucleotide) | Trilink Biotechnologies | | |
| sequence-based reagent | AAU (oligoribonucleotide) | Trilink Biotechnologies | | |
| sequence-based reagent | JW 30 (oligodeoxyribonucleotide) | Integrated DNA Technologies | | |
| sequence-based reagent | JW 61 (oligodeoxyribonucleotide) | Integrated DNA Technologies | | |
| sequence-based reagent | JW 62 (oligodeoxyribonucleotide) | Integrated DNA Technologies | | |
| sequence-based reagent | JW 85 (oligodeoxyribonucleotide) | Integrated DNA Technologies | | |
| sequence-based reagent | LY10 (oligodeoxyribonucleotide) | Integrated DNA Technologies | | |
| sequence-based reagent | LY11 (oligodeoxyribonucleotide) | Integrated DNA Technologies | | |
| sequence-based reagent | S128a (oligodeoxyribonucleotide) | Integrated DNA Technologies | | |
| sequence-based reagent | S1219 (oligodeoxyribonucleotide) | Integrated DNA Technologies | | |
| sequence-based reagent | S1220 (oligodeoxyribonucleotide) | Integrated DNA Technologies | | |
| sequence-based reagent | RPOC820 (oligodeoxyribonucleotide) | Integrated DNA Technologies | | |
| sequence-based reagent | RPOC3140 (oligodeoxyribonucleotide) | Integrated DNA Technologies | | |
| sequence-based reagent | SbfRPOC50 (oligodeoxyribonucleotide) | Integrated DNA Technologies | | |
| sequence-based reagent | Taq_rpoC_F (oligodeoxyribonucleotide) | Integrated DNA Technologies | | |
| sequence-based reagent | Taq_rpoC_R (oligodeoxyribonucleotide) | Integrated DNA Technologies | | |
| sequence-based reagent | XbaRPOC4050 (oligodeoxyribonucleotide) | Integrated DNA Technologies | | |
| peptide, recombinant protein (E.coli) | RNAP polymerase core enzyme | PMID: 12511572 | | |
| peptide, recombinant protein (E.coli) | sigma70 | PMID: 9157885 | | |
| peptide, recombinant protein (E.coli) | RNAP core enzyme derivative beta'R1148Bpa | PMID: 26257284 | | |
| peptide, recombinant protein (E.coli) | RNAP core enzyme derivative beta'T48Bpa | PMID: 26257284 | | |
| peptide, recombinant protein (E.coli) | RNAP core enzyme derivative beta'R1148Bpa;D460A | this work | | |
| peptide, recombinant protein (E.coli) | RNAP core enzyme derivative beta'R1148Bpa;D460A | this work | | |
| peptide, recombinant protein | Dam methyltransferase | New England BioLabs | Cat# M0222S | |
| peptide, recombinant protein | restriction endonuclease SbfI-HF | New England BioLabs | Cat# R3642S | |
| peptide, recombinant protein | restriction endonuclease XbaI | New England BioLabs | Cat# R0145S | |
| peptide, recombinant protein | T4 polynucleotide kinase | New England BioLabs | Cat# M0201S | |
| antibody | anti-digoxigenin antibody | Sigma-Aldrich | Cat# 11333089001 | |
| commercial assay or kit | Dynabeads MyOne Streptavidin C1 | Thermo-Fisher | Cat# 650.01 | |
| commercial assay or kit | MagneHis protein purification system | Promega | Cat# V8500 | |
| commercial assay or kit | TRI reagent | Molecular Research Center | Cat# TR118 | |
| chemical compound, drug | carbenicillin | Gold Biotechnology | cat# c-103-25 | |
| chemical compound, drug | chloramphenicol | Gold Biotechnology | cat# c-105-25 | |

*Continued*

| Reagent type (species) or resource | Designation | Source or reference | Identifiers | Additional information |
|---|---|---|---|---|
| chemical compound, drug | spectinomycin | Gold Biotechnology | cat# 22189-32-8 | |
| chemical compound, drug | streptomycin | Omnipur | cat# 3810-74-0 | |
| software, algorithm | SigmaPlot | Systat Software | | |
| software, algorithm | xvin | PMID: 16156080 | | |

## Proteins

Wild-type *E. coli* RNAP core enzyme used in transcription experiments was prepared from *E. coli* strain NiCo21(DE3) (New England BioLabs) transformed with plasmid pIA900 (*Svetlov and Artsimovitch, 2015*) as described (*Winkelman et al., 2015*). Wild-type RNAP for single-molecule DNA-nanomanipulation experiments was prepared from *E. coli* strain BL21(DE3) (New England Biolabs) transformed with plasmid pVS10 (*Artsimovitch et al., 2003*) as described (*Artsimovitch et al., 2003*).

Bpa-containing RNAP core-enzyme derivatives for in vitro protein-DNA photocrosslinking (β′R1148Bpa for analysis of RNAP leading-edge positions; β′T48Bpa for analysis of RNAP trailing-edge positions) were prepared from *E. coli* strain NiCo21(DE3) (New England BioLabs) co-transformed with plasmid pEVOL-pBpF (*Chin et al., 2002*; Addgene) and plasmid pIA900-β′R1148Bpa (*Winkelman et al., 2015*) or pIA900-β′T48Bpa (*Winkelman et al., 2015*), as in *Winkelman et al. (2015)*.

Bpa-containing, mutationally inactivated, RNAP core-enzyme derivatives for in vitro and in vivo protein-DNA photocrosslinking (β′R1148Bpa;β′D460A for analysis of RNAP leading-edge positions; β′T48Bpa;β′D460A for analysis of RNAP trailing-edge positions) were prepared from *E. coli* strain NiCo21(DE3) (New England BioLabs) co-transformed with plasmid pEVOL-pBpF (*Chin et al., 2002*; Addgene) and plasmid pIA900-β′R1148Bpa;β′D460A or pIA900-β′T48Bpa;β′D460A [constructed from pIA900-β′R1148-Bpa (*Winkelman et al., 2015*) and pIA900-β′T48-Bpa (*Winkelman et al., 2015*) by use of site-directed mutagenesis with primer 'JW30', as in *Winkelman et al. (2015)*].

σ$^{70}$ was prepared from *E. coli* strain BL21(DE3) (New England Biolabs) transformed with plasmid pσ$^{70}$-His (*Marr and Roberts, 1997*) as described (*Marr and Roberts, 1997*). To form RNAP holoenzyme, 1 μM RNAP core enzyme and 5 μM σ$^{70}$ in 10 mM Tris-Cl, pH 8.0, 100 mM KCl, 10 mM MgCl$_2$, 0.1 mM EDTA, 1 mM DTT, and 50% glycerol were incubated 30 min at 25°C.

## Oligonucleotides

Oligodeoxyribonucleotides (desalted) were purchased from IDT (sequences in *Supplementary file 1*). Oligoribonucleotides (HPLC-purified) were purchased from Trilink Biotechnologies.

## Determination of TSS position in vitro

Experiments in *Figure 1B* and *Figure 1—figure supplement 2B* were performed using reaction mixtures (60 μl) containing 20 nM RNAP holoenzyme derivative, 4 nM plasmid pCDF-CP-*lac*CONS-GGG or pCDF-CP-*lac*CONS-CCT, carrying derivatives of the *lac*CONS promoter (*Mukhopadhyay et al., 2001*) having a GGG or CCT discriminator [prepared by inserting a synthetic 248 bp DNA fragment (5′- GAAGCCCTGCATTAGGGGTACCCTAGAGCCTGACCGGCATTATAGCCCCAGCGGCGGA TCCCTGCGGGTCGACAAGCTTGAATAGCCATCCCAATCGAACAGGCCTGCTGGTAA TCGCAGGCCTTTTTATTTGGATGGAGCTCTGAGAGTCTTCGGTGTATGGGTTTTGCGGTGGAAA- CACAGAAAAAAGCCCGCACCTGACAGTGCGGGCTTTTTTTTTCGACCAAAGGGACGACCGGG TCGTTGGT- 3′) between positions 3601 and 460 of pCDF-1b (EMD-Millipore), yielding plasmid pCDF-CP, followed by ligating a 200 bp BglII-digested DNA fragment carrying the *lac*CONS promoter with GGG or CCT discriminator (5′-GTTCAGAGTTCTACAGTCCGACGATCGCGGATGC TTGACAGAGTGAGCGCAACGCAATAACAGTCATCTAGATAGAACTTTAGGCACCCCAGGCTTGA- CACTTTATGCTTCGGCTCGTATAATGGGGATGCATGTGAGCGGATAACAATGCGGTTAGGCTTA- GAGCGCTTAGTCGATGCTGGAATTCTCGGGTGCCAAGG−3′ or 5′-GTTCAGAGTTCTACAG TCCGACGATCGCGGATGCTTGACAGAGTGAGCGCAACGCAATAACAGTCATCTAGATAGAAC

TTTAGGCACCCCAGGCTTGACACTTTATGCTTCGGCTCGTATAATCCTGATGCATGTGAGCGGA TAACAATGCGGTTAGGCTTAGAGCGCTTAGTCGATGCTGGAATTCTCGGGTGCCAAGG−3';  −35 and −10 elements underlined; discriminator in bold) with BglI-digested plasmid pCDF-CP], 0 or 1 mM ATP, 0 or 1 mM CTP, 0 or 1 mM GTP, and 0 or 1 mM UTP in 60 µl 10 mM Tris-Cl, pH 8.0, 70 mM NaCl, 10 mM MgCl$_2$, and 0.1 mg/ml bovine serum albumin. After 20 min at 37°C, reactions were terminated by addition of 100 µl 10 mM EDTA pH 8.0 and 1 mg/ml glycogen. Samples were extracted with acid phenol:chloroform (*Sambrook and Russell, 2001*) (Ambion), and RNA products were recovered by ethanol precipitation (*Sambrook and Russell, 2001*) and re-suspended in 6.5 µl water. The RNA products were analyzed by primer extension to define TSS positions. Primer-extension was performed by combining 6.5 µl RNA products in water, 1 µl 1 µM $^{32}$P-5'-end-labeled primer 's128a' [*Supplementary file 1*; 200 Bq/fmol; prepared using [γ$^{32}$P]-ATP (PerkinElmer) and T4 polynucleotide kinase (New England Biolabs); procedures as in *Sambrook and Russell, 2001*], and 1 µl 10x avian myelobastosis virus (AMV) reverse transcriptase buffer (New England BioLabs) heating 10 min at 90°C, cooling to 40°C at 0.1 °C/s, and incubating 15 min at 40°C; adding 0.5 µl 10 mM dNTP mix (2.5 mM dATP, 2.5 mM dGTP, 2.5 mM, dCTP, and 2.5 mM dTTP; New England Biolabs) and 1 µl 10 U/µl AMV reverse transcriptase (New England BioLabs); and incubating 1 hr at 50°C. Primer-extension reactions were terminated by heating 20 min at 85°C; 10 µl 1x TBE (*Sambrook and Russell, 2001*), 8 M urea, 0.025% xylene cyanol, and 0.025% bromophenol blue was added; and samples were analyzed by electrophoresis on 8 M urea, 1X TBE polyacrylamide gels UreaGel System; National Diagnostics) (procedures as in *Sambrook and Russell, 2001*), followed by storage-phosphor imaging (Typhoon 9400 variable-mode imager; GE Life Science). TSS positions were determined by comparison to products of a DNA-nucleotide sequencing reaction obtained using a PCR-generated DNA fragment containing positions −129 to +71 of the *lac*CONS-GGG promoter and primer 's128a' (Thermo Sequenase Cycle Sequencing Kit; Affymetrix; methods as per manufacturer). Experiments in *Figure 3B*, were performed analogously, but using a 1.3 kb DNA fragment carrying positions −687 to +644 of the *lac*CONS promoter (*Mukhopadhyay et al., 2001*) prepared by PCR amplification of plasmid pUC18-T20C2-lac*CONS* [prepared by replacing the SbfI-XbaI segment of plasmid pUC18 (Thermo Scientific) with a 2.0 kb SbfI-XbaI DNA fragment obtained by PCR amplification of *Thermus aquaticus rpoC* gene with primers Taq_rpoC_F and Taq_rpoC_R (*Supplementary file 1*) and digestion with XbaI and SbfI-HF (New England BioLabs), yielding plasmid pUC18-T20C2, followed by inserting a synthetic 117 bp DNA fragment carrying the *lac*CONS promoter (5'-CGGATGCTTGACAGAGTGAGCGCAACGCAATAACAGTCATCTAGATAGAAC TTTAGGCACCCCAGGCTTGACACTTTATGCTTCGGCTCGTATAATGTGTGGAATTGTGAGCGGA TA-3'; −35 and −10 elements underlined; discriminator in bold) into the KpnI site of plasmid pUC18-T20C2] with primers 'LY10' and 'LY11' (*Supplementary file 1*), and performing experiments in the presence of 0 or 1 mM UGG, GGA, GAA, or AAU.

## Determination of TSS position in vivo

*E. coli* strain NiCo21(DE3) (New England BioLabs) transformed with plasmid pCDF-CP-*lac*CONS-GGG or pCDF-CP-*lac*CONS-CCT was plated on LB agar (*Sambrook and Russell, 2001*) containing 50 µg/ml spectinomycin and 50 µg/ml streptomycin, single colonies were inoculated into 25 ml LB broth (*Sambrook and Russell, 2001*) containing 50 µg/ml spectinomycin and 50 µg/ml streptomycin in 125 ml Bellco flasks, and cultures were shaken (220 rpm) at 37°C. When cell densities reached OD$_{600}$ = 0.6, 2 ml aliquots were centrifuged 2 min at 4°C at 23,000xg, and resulting cell pellets were frozen at −80°C. Cell pellets were thawed in 1 ml TRI Reagent (Molecular Research Center) at 25°C for 5 min, completely re-suspended by pipetting up and down, incubated 10 min at 70°C, and centrifuged 2 min at 25°C at 23,000 x g. Supernatants were transferred to fresh 1.7 ml microfuge tubes, 200 µl chloroform (Ambion) was added, vortexed, and samples were centrifuged 1 min at 25°C at 23,000 x g. Aqueous phases were transferred to a fresh tube and nucleic acids were extracted with acid phenol:chloroform (*Sambrook and Russell, 2001*). Nucleic acids were recovered by ethanol precipitation (*Sambrook and Russell, 2001*), and re-suspended in 20 µl 10 mM Tris-Cl, pH 8.0. Primer extension was performed as described in the preceding section.

## Determination of RNAP leading-edge and trailing-edge positions in vitro: protein-DNA photocrosslinking in vitro

Experiments in *Figure 1B* were performed using reaction mixtures (10 µl) containing 50 nM Bpa-containing RNAP holoenzyme derivative β'R1148Bpa (for analysis of RNAP leading-edge positions) or β'T48Bpa (for analysis of RNAP trailing-edge positions) and 4 nM plasmid pCDF-CP-*lac*CONS-GGG or plasmid pCDF-CP-*lac*CONS-CCT in 10 mM Tris-Cl, pH 8.0, 70 mM NaCl, 10 mM MgCl$_2$, and 0.1 mg/ml bovine serum albumin. Reaction mixtures were incubated 5 min at 37°C, UV-irradiated 5 min at 25°C in a Rayonet RPR-100 photochemical reactor equipped with 16 × 350 nm tubes (Southern New England Ultraviolet), and resulting protein-DNA crosslinks were mapped using primer extension. Primer-extension reactions (12.5 µl) were performed by combining 2 µl aliquot of crosslinking reaction, 1 µl 1 µM $^{32}$P-5'-end-labeled primer 's128a' (for analysis of leading-edge positions) or primer 'JW85' (for analysis of trailing-edge positions) [*Supplementary file 1*; 200 Bq/fmol; prepared using [γ$^{32}$P]-ATP (PerkinElmer) and T4 polynucleotide kinase (New England Biolabs); procedures as in *Sambrook and Russell, 2001*, 1 µl 10X dNTPs (2.5 mM dATP, 2.5 mM dCTP, 2.5 mM dGTP, 2.5 mM TTP, 0.5 µl 5 U/µl Taq DNA polymerase (New England BioLabs), 5 µl 5 M betaine, 0.625 µl 100% dimethyl sulfoxide, and 1.25 µl 10x Taq DNA polymerase buffer (New England BioLabs); and cycling 16–40 times through 30 s at 95°C, 30 s at 53°C, and 30 s at 72°C. Primer-extension reactions were terminated, and primer-extension products were analyzed as in the preceding section. Experiments in *Figure 1—figure supplement 2B* were performed analogously, but using Bpa-containing, mutationally inactivated, RNAP derivatives β'R1148Bpa; β'D460A (for analysis of RNAP leading-edge positions) and β'T48Bpa; β'D460A (for analysis of RNAP trailing-edge positions)

Experiments in *Figure 3B* and *Figure 3—figure supplement 1B* were performed analogously, but using reaction mixtures also containing 0 or 1 mM of ribotrinucleotide primers UGG, GGA, GAA, or AAU, and using $^{32}$P-5'-end-labeled primers 'JW62' and 'JW61' (*Supplementary file 1*) in primer-extension reactions.

## Determination of RNAP leading-edge and trailing-edge positions in vivo: protein-DNA photocrosslinking in vivo

Experiments in *Figure 1B* were performed using a three-plasmid merodiploid system that enabled production of a Bpa-containing, mutationally inactivated, decahistidine-tagged RNAP holoenzyme derivative in vivo and enabled trapping of RPo consisting of the Bpa-containing, mutationally inactivated, decahistidine-tagged RNAP holoenzyme derivative and a *lac*CONS promoter with GGG or CCT discriminator in vivo, and UV-irradiation of cells (*Figure 1—figure supplement 1*).

*E. coli* strain NiCo21(DE3) (New England BioLabs) transformed sequentially with (i) plasmid pCDF-CP-*lac*CONS-GGG or plasmid pCDF-CP-*lac*CONS-CCT, (ii) plasmid pIA900-β'T48Bpa; β'D460A or plasmid pIA900-β'R1148Bpa; β'D460A, and (iii) plasmid pEVOL-pBpF (*Chin et al., 2002*; Addgene) was plated to yield a confluent lawn on LB agar (*Sambrook and Russell, 2001*) containing 100 µg/ml carbenicillin, 50 µg/ml spectinomycin, 50 µg/ml streptomycin, and 25 µg/ml chloramphenicol; cells were scraped from the plate and used to inoculate 250 ml LB broth (as described above) containing 1 mM Bpa (Bachem), 100 µg/ml carbenicillin, 50 µg/ml spectinomycin, 50 µg/ml streptomycin, and 25 µg/ml chloramphenicol in a 1000 ml flask (Bellco) to yield OD$_{600}$ = 0.3; the culture was shaken (220 rpm) 1 hr at 37°C in the dark, isopropyl-β-D-thiogalactoside was added to 1 mM; and the culture was further shaken (220 rpm) 3 hr at 37°C in the dark. Aliquots (7 ml) were transferred to 13 mm x 100 mm borosilicate glass test tubes (VWR), UV-irradiated 20 min at 25°C in a Rayonet RPR-100 photochemical reactor equipped with 16 × 350 nm tubes (Southern New England Ultraviolet), harvested by centrifuging 15 min at 4°C at 3000xg, and cell pellets were frozen at −20°C. Cell pellets were thawed 30 min at 4°C, re-suspended in 40 ml 50 mM Na$_2$HPO$_4$ pH 8.0, 1.4 M NaCl, 20 mM imidazole, 14 mM β-mercaptoethanol, 0.1% Tween20, and 5% ethanol containing 2 mg egg white lysozyme. Cells were lysed by sonication 5 min at 4°C., cell lysates were centrifuged 40 min at 4°C at 23,000xg, and supernatants were added to 1 ml Ni-NTA-agarose (Qiagen, Germantown, MD) in 1 ml 50 mM Na$_2$HPO$_4$, pH 8.0, 1.4 M NaCl, 20 mM imidazole, 0.1% Tween-20, 5 mM β-mercaptoethanol, and 5% ethanol, and incubated 30 min at 4°C with gentle rocking. The Ni-NTA-agarose was loaded into a 15 ml polyprep column (BioRad), the resulting column was washed with 10 ml of 50 mM Na$_2$HPO$_4$, pH 8.0, 300 mM NaCl, 20 mM imidazole, 0.1% Tween-20, 5 mM β-mercaptoethanol, and 5% ethanol and eluted with 3 ml of the same buffer containing 300 mM imidazole. The

eluate was concentrated to 0.2 ml using an 1000 MWCO Amicon Ultra-4 centrifugal filter (EMD Millipore); the buffer was exchanged to 0.2 ml 20 mM Tris-Cl, pH 8.0, 200 mM KCl, 20 mM MgCl$_2$, 0.2 mM EDTA, and 1 mM DTT using the 1000 MWCO Amicon Ultra-4 centrifugal filter (EMD Millipore); 0.2 ml glycerol was added; and the sample was stored at −20°C. Protein-DNA crosslinks were mapped by denaturation followed by primer extension. Denaturation was performed by combining 25 μl crosslinked RNAP-DNA, 25 μl water, 15 μl 5 M NaCl, and 6 μl 100 μg/ml heparin; heating 5 min at 95°C; cooling on ice. Denatured crosslinked RNAP-DNA was purified by adding 20 μl Magne-His Ni-particles (Promega) equilibrated and suspended in 10 mM Tris-Cl, pH 8.0, 1.2 M NaCl, 10 mM MgCl$_2$, 10 μg/ml heparin, and 0.1 mg/ml bovine serum albumin; washing once with 50 μl 10 mM Tris-Cl, pH 8.0, 1.2 M NaCl, 10 mM MgCl$_2$, 10 μg/ml heparin, and 0.1 mg/ml bovine serum albumin; washed twice with 50 μl 1x *Taq* DNA polymerase buffer (New England BioLabs); and resuspended in 10 μl 1x *Taq* DNA polymerase buffer. Primer extension was performed using 2 μl aliquots of purified denatured crosslinked RNAP-DNA, using procedures essentially as described above for experiments in *Figure 1B*.

## Determination of RNAP-dependent DNA unwinding by single-molecule DNA-nanomanipulation: DNA constructs

2.0 kb DNA fragments carrying single centrally located *lac*CONS-GGG, *lac*CONS-CCT, or *lac*CONS promoters were prepared by digesting plasmid pUC18-T20C2-*lac*CONS-GGG or plasmid pUC18-T20C2-*lac*CONS-CCT [prepared by inserting a synthetic 80 bp DNA fragment carrying a derivative of the *lac*CONS promoter (*Mukhopadhyay et al., 2001*) having a GGG or CCT discriminator (5'-CATCTAGATCACATTTTAGGCACCCCAGGCTTGACACTTTATGCTTCGGCTCGTATAATGGGGATGCATGTGAGCGGATA-3' or 5'-CATCTAGATCACATTTTAGGCACCCCAGGCTTGACACTTTATGCTTCGGCTCGTATAATCCTGATGCATGTGAGCGGATA −3'; −35 and −10 elements underlined; discriminator in bold) into the KpnI site of plasmid pUC18-T20C2] or plasmid pUC18-T20C2-*lac*CONS with XbaI and SbfI-HF (New England BioLabs), followed by agarose gel electrophoresis.

1.3 kb DNA fragments carrying a single centrally located *lac*CONS promoter were prepared by PCR amplification of plasmid pUC18-T20C2-*lac*CONS, using primers 'LY10' and 'LY11' (*Supplementary file 1*), followed by treatment with Dam methyltransferase (New England BioLabs), digestion with XbaI and SbfI-HF (New England BioLabs), and agarose gel electrophoresis.

DNA constructs for magnetic-tweezers single-molecule DNA-nanomanipulation were prepared from the above 2.0 kb and 1.3 kb DNA fragments by ligating, at the XbaI end, a 1.0 kb DNA fragment bearing multiple biotin residues on both strands [prepared by PCR amplification of plasmid pARTaqRPOC-*lac*CONS using primers 'XbaRPOC4050' and 'RPOC3140' *Supplementary file 1*) and conditions as described (*Revyakin et al., 2004*, *2005*, *2006*, Revyakin et al., 2003

## Determination of RNAP-dependent DNA unwinding by single-molecule DNA-nanomanipulation: data collection

Experiments were performed essentially as described (*Revyakin et al., 2004*, *2005*, *2006*, Revyakin et al., 2003

Experiments in *Figure 2* (experiments addressing TSS selection for promoters with GGG or CCT discriminator sequence), were performed using standard reactions containing mechanically extended, torsionally constrained, 2.0 kb DNA molecule carrying GGG or CCT promoter (extension force = 0.3 pN; superhelical density = 0.021 for experiments with positively supercoiled DNA; superhelical density = −0.021 for experiments with negatively supercoiled DNA) and RNAP holoenzyme (10 nM for experiments with positively supercoiled DNA; 0.5 nM for experiments with negatively supercoiled DNA) in 25 mM Na-HEPES, pH 7.9, 75 mM NaCl, 10 mM MgCl$_2$, 1 mM dithiothreitol, 0.1% Tween-20, 0.1 mg/ml bovine serum albumin) at 30°C. Data from each of three single DNA molecules were pooled [differences in plectoneme size (at superhelical density ± 0.021) and $\bar{\Delta l}_{obs} \leq 5\%$].

Experiments in *Figure 4—figure supplement 1* (experiments demonstrating that reduction in DNA-fragment length from 2.0 to 1.3 kb enables single-bp resolution) were performed using standard reactions containing mechanically extended, torsionally constrained, 2.0 kb or 1.3 kb DNA molecule carrying *lac*CONS promoter (extension force = 0.3 pN; initial superhelical density = 0.021 or 0.024 for experiments with 2.0 kb or 1.3 kb positively supercoiled DNA; superhelical density = −0.021 or −0.024 for experiments with 2.0 kb or 1.3 kb negatively supercoiled DNA) and

RNAP holoenzyme (10 nM for experiments with positively supercoiled DNA; 0.5 nM for experiments with negatively supercoiled DNA) in the buffer of the preceding paragraph at 30°C. For each DNA-fragment length, data were collected on one single DNA molecule.

Experiments in *Figure 4* and *Figure 4—figure supplement 2* (experiments addressing primer-programmed TSS selection with primers UGG, GGA, GAA, AAU) were performed using standard reactions containing mechanically extended, torsionally constrained 1.3 kb DNA molecule carrying *lac*CONS promoter (extension force = 0.3 pN; initial superhelical density = 0.024 for experiments with positively supercoiled DNA; superhelical density = −0.024 for experiments with negatively supercoiled DNA) and RNAP holoenzyme (10 nM for experiments with positively supercoiled DNA; 0.5 nM for experiments with negatively supercoiled DNA) in the buffer of the preceding paragraph at 30°C. Primers UGG, GGA, GAA, and AAU were present at 0 or 1 mM, 0 or 1 μM, 0 or 2.5 μM, and 0 or 1 mM, respectively. For experiments with positively supercoiled DNA, data from each of seven single DNA molecules were normalized based on $\Delta\bar{l}_{obs,pos}$ in absence of primer and pooled; for experiments with negatively supercoiled DNA, data from each of two single DNA molecules were normalized based on $\Delta\bar{l}_{obs,neg}$ in absence of primer and pooled.

Experiments in *Figures 5–6* and *Figure 6—figure supplement 1* (experiments addressing primer-concentration dependences of $t_{unwound}$ in primer-programmed TSS selection) were performed using standard reactions containing mechanically extended, torsionally constrained, 2.0 kb DNA molecule carrying *lac*CONS promoter (extension force = 0.3 pN; initial superhelical density = 0.021) and RNAP holoenzyme (10 nM) in the buffer of the preceding paragraph at 30°C. Each titration consisted of recordings in absence of primer followed by recordings in presence of primer at increasing concentrations. (0, 0.50, 0.90, 1.3, 250, 500, 750, and 1000 μM for UGG; 0, 0.25, 0.50, and 1.0 μM for GGA; 0, 0.50, 1.0, and 1.5 μM for GAA; 0, 0.40, 0.80, 1.2, 200, 400, 600, and 800 μM for AAU). For each titration, data were collected on one single DNA molecule.

For experiments with negatively supercoiled DNA, for which $\bar{t}_{unwound} \gg 1$ h (*Revyakin et al., 2004*, *2005*, *2006*, Revyakin et al., 2003

## Determination of RNAP-dependent DNA unwinding by single-molecule DNA-nanomanipulation: data reduction for determination of DNA unwinding

Raw time traces were analyzed to yield DNA extension (*l*) as described (*Revyakin et al., 2004*, *2005*, *2006*, Revyakin et al., 2003

Changes in *l* attributable to DNA unwinding ($\Delta l_u$) and changes in *l* attributable to DNA compaction ($\Delta l_c$) were calculated as: $\Delta l_u = (\Delta l_{obs,neg} + \Delta l_{obs,pos})/2$, and $\Delta l_c = (\Delta l_{obs,pos} - \Delta l_{obs,neg})/2$, where $\Delta l_{obs,pos}$ and $\Delta l_{obs,neg}$ are observed changes in *l* in experiments with positively supercoiled DNA and negatively supercoiled DNA, as described (*Revyakin et al., 2004*, *2005*, *2006*, Revyakin et al., 2003

## Determination of RNAP-dependent DNA unwinding by single-molecule DNA-nanomanipulation: data reduction for determination of energetics of scrunching and anti-scrunching

Lifetimes of unwound states ($t_{unwound}$) were extracted from single-molecule traces as described (*Revyakin et al., 2004*, *2005*, *2006*, Revyakin et al., 2003

For experiments in absence of primer (*Qi and Turnbough, 1995*; *Figure 6—figure supplement 1*):

$$\mathrm{R} + \mathrm{P} \underset{}{\overset{\mathrm{K_B}}{\rightleftharpoons}} \mathrm{RPc} \underset{\mathrm{k_{-2}}}{\overset{\mathrm{k_2}}{\rightleftharpoons}} \mathrm{RPo}$$

where R, P, RPc, and RPo denote RNAP holoenzyme, promoter, RNAP-promoter closed complex, and RNAP-promoter open complex; and

$$\bar{t}_{unwound} = \frac{1}{k_{-2}}$$

For experiments in presence of primer GGA, which programs TSS selection at modal position and therefore does not require scrunching or anti-scrunching for TSS selection (*Figure 6—figure supplement 1*):

$$R + P + NpNpN \overset{K_B}{\rightleftharpoons} RPc + NpNpN \overset{k_2}{\underset{k_{-2}}{\rightleftharpoons}} RPo + NpNpN \overset{K_{NpNpN}}{\rightleftharpoons} RPo : NpNpN$$

where NpNpN denotes primer; and

$$\bar{t}_{unwound} = \frac{K_{NpNpN}}{k_{-2}} * [NpNpN] + \frac{1}{k_{-2}}$$

For experiments in presence of primer UGG, or GAA, or AAU, which program TSS selection at positions different form modal TSS, and therefore require scrunching or anti-scrunching for TSS selection (*Figures 4–6* and *Figure 6—figure supplement 1*):

$$R + P + NpNpN \overset{K_B}{\rightleftharpoons} RPc + NpNpN \overset{k_2}{\underset{k_{-2}}{\rightleftharpoons}} RPo + NpNpN \overset{K_{scrunch}}{\rightleftharpoons} RPo' + NpNpN \overset{K_{NpNpN}}{\rightleftharpoons} RPo' : NpNpN$$

where RPo' denotes a scrunched or anti-scrunched RPo; and

$$\bar{t}_{unwound} = \frac{K_{NpNpN} * K_{scrunch}}{k_{-2}} * [NpNpN] + \frac{1 + K_{scrunch}}{k_{-2}}$$

$K_{scrunch}$, $K_{NpNpN}$, $\Delta G_{NpNpN}$, and $\Delta G_{scrunch}$ were obtained by fitting slopes and y-intercepts of linear-regression fits of plots of $\bar{t}_{unwound}$ vs. primer concentration (*Figure 6D* and *Figure 6—figure supplement 1*) to the equation of *Figure 6C*, stipulating $K_{scrunch} = 1$ and $\Delta G_{scrunch} = 0$ for primer GGA, which programs TSS selection at modal position and therefore does not require scrunching or anti-scrunching for TSS selection.

## Quantitation and statistical analysis
Data in *Figure 2* are means ± SEM of at least 70 technical replicates of each of three biological replicates (three single DNA molecules) for positively supercoiled DNA and at least 50 technical replicates of each of three biological replicates (three single DNA molecules) for negatively supercoiled DNA.

Data in *Figure 4—figure supplement 1A–B* are means ± SEM of at least 100 technical replicates for a single DNA molecule (positively supercoiled DNA) or at least 70 technical replicates for a single DNA molecule (negatively supercoiled DNA).

Data in *Figure 4—figure supplement 1C* are means ± SEM of randomly selected subsets of n = 30. Similar results were obtained for ten different randomly selected subsets of n = 30.

Data in *Figure 4B–C* and *Figure 4—figure supplement 2* are means ± SEM of at least 40 technical replicates for each of seven biological replicates (seven single DNA molecules) for positively supercoiled DNA and at least 50 technical replicates for each of two biological replicates (two single DNA molecules) for negatively supercoiled DNA.

Data in *Figures 5–6* and *Figure 6—figure supplement 1* are means ± SEM of at least 150 technical replicates for one single DNA molecule for each of the four primers.

## Acknowledgements
Work was supported by NIH grants GM041376 (RHE), GM118059 (BEN), and a European Science Foundation EURYI grant (TRS). We thank Seth Goldman for help with construction of pCDF-CP.

## Additional information

### Funding

| Funder | Grant reference number | Author |
| --- | --- | --- |
| National Institutes of Health | GM041376 | Richard H Ebright |
| National Institutes of Health | GM118059 | Bryce E Nickels |
| European Science Foundation | EURYI | Terence Strick |

The funders had no role in study design, data collection and interpretation, or the decision to submit the work for publication.

## Author contributions

Libing Yu, Jared T Winkelman, Conceptualization, Data curation, Formal analysis, Investigation, Methodology, Writing—review and editing; Chirangini Pukhrambam, Resources, Investigation; Terence R Strick, Conceptualization, Funding acquisition, Methodology, Writing—review and editing; Bryce E Nickels, Richard H Ebright, Conceptualization, Resources, Supervision, Funding acquisition, Writing—original draft, Project administration, Writing—review and editing

## Author ORCIDs

Richard H Ebright 🔟 https://orcid.org/0000-0001-8915-7140

## Decision letter and Author response

Decision letter https://doi.org/10.7554/eLife.32038.018
Author response https://doi.org/10.7554/eLife.32038.019

## Additional files

### Supplementary files

• Supplementary file 1. Oligonucleotides
DOI: https://doi.org/10.7554/eLife.32038.015

• Transparent reporting form
DOI: https://doi.org/10.7554/eLife.32038.016

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
