## [Decision Letter]

Thank you for submitting your article "The mechanism of transcription start site selection" for consideration by *eLife*. Your article has been reviewed by three peer reviewers, one of whom is a member of our Board of Reviewing Editors, and the evaluation has been overseen by Kevin Struhl as the Senior Editor. The reviewers have opted to remain anonymous.

As you can see from the comments below, the reviewers agree that the work merits publication and is clearly written, adds to the mechanistic understanding of how the chosen transcription start site modulates scrunching and antiscrunching behavior, both in vitro and in vivo. I ask you to revise the manuscript in response to the comments and suggestions. Because of the nature of the comments, I do not believe additional experiments are needed.

*Reviewer #1:*

In previous studies, Ebright and Nickels determined TSS and positions of leading and trailing edges of *E. coli* RNA polymerase from randomly mutagenized library of promotor sequences. They found a strong correlation between TSS and leading edge position, not between TSS and trailing edge, leading to the proposal of scrunching and antiscrunching for TSS selection downstream and upstream of canonical TSS, respectively. There are three main results reported here.

1) They developed an in vivo crosslinking to demonstrate that the leading edge movement seen in vitro is also seen in vivo.

2) They used magnetic tweezers to confirm that DNA unwinding is beyond the canonical amount for scrunching and vice versa. Importantly, they could show the change in DNA unwinding amount precisely corresponds to the distance by which TSS moves.

3) Finally, they determined the lifetime of the triplet primer bound state to canonical, scrunched and anti-scrunched transcription bubble as a function of primer concentration, allowing them to estimate the energetic cost of scrunching and anti-scrunching. Satisfyingly, the energetic cost they determined can explain the observed distribution of TSS from the library they studied previously.

Generally, the work is of high quality and provides additional validation to their prior proposals. There is no surprise or additional insight arising. The third component of this work provides a potentially important set of parameters, but what's unsatisfying is why certain sequences move TSS upstream vs downstream. Can the authors use the known sequence-dependent physical chemistry properties of DNA to predict which direction TSS moves?

*Reviewer #2:*

Based on in vitro experiments, Ebright and Nichols have previously shown that the variability in the distance of the transcription start site (TSS) with respect to core promoter elements occurs through DNA scrunching and anti-scrunching such that the position of the RNAP trailing edge remains fixed, while the leading edge moves on the DNA with concomitant expansion or contraction of the transcription bubble when the RNAP open complex (RPo) is made, and is dependent on the discriminator sequence. In this manuscript, they expand the observations previously published in 2016 by exploring (1) the effect of determinants of TSS selection in vivo, (2) required changes in the transcription bubble in vitro, (3) the source of energy that elicits scrunching and anti-scrunching since, in contrast to this activity in initiation, there is no RNA synthesis. This manuscript, characterized by the development and use of cutting edge approaches, provides answers to important questions in the process of transcription initiation.

To determine the dynamics of RNAP through TSS changes in vivo, they developed an approach to analyze the RNAP open complex (RPo) in cells. To that end, cells contained three plasmids: (1) carrying the promoter under study, (2) encoding a Bpa-specific nonsense suppressor tRNA and its cognate charging synthase, (3) a ß' gene with (i) nonsense mutations where Bpa is incorporated to mark the sites of the RPo leading and training edges by UV crosslinking to the DNA, (ii) a decahistidine-tag for complex isolation and analysis, and (iii) the ß'D460A mutation to abrogate the binding of Mg^2+^ and NTPs at the active site to trap RPo in vivo. Cells containing the wildtype ß' subunit were grown in media containing Bpa. in vivo and in vitro TSS and leading and trailing edges were identical indicating that scrunching occurs in vivo and is dependent on the discriminator sequence. Changes in TSS and in leading edges implicate scrunching and concomitant changes in the extent of the transcription bubble, changes that were measured by DNA unwinding to single-nucleotide base pair and sub-second temporal resolution using magnetic-tweezers DNA nano-manipulation under positive and negative supercoiling as developed by Strick.

To overcome the restriction in TSS provided by the GGG and CTT discriminator sequences, they expanded the range of sites of initiation by taking advantage of primer-programmed TSS selection with primers that determine upstream and downstream initiation from the modal TSS position. As predicted, primer initiation upstream of the modal site involves anti-scrunching. Using measurements of the life times of the unwound state as a function of primer concentration allowed the determination of energy costs and probability of use of the different TSSs. The results indicate that TSS initiation does not require energy beyond that provided by the thermal bath and explain why initiation upstream or downstream of the modal is disfavored.

Subsection “TSS selection exhibits first hallmark of scrunching – movements of RNAP leading edge but not RNAP trailing edge – both in vitro and in vivo” from bottom carrying genes for *an* engineered.

*Reviewer #3:*

This paper reports in vitro and in vivo analyses of the amount of scrunching within the open complex of bacterial RNA polymerase and promoters with various distances between the -10 element and the start of transcription +1 (TSS). The authors employ BpA crosslinking between residues within RNAP and the DNA to demonstrate that the back end of the polymerase stays fixed while the front end moves forward when the length between the -10 element and the TSS increases. Using a magnetic-tweezers single-molecule DNA-nanomanipulation assay, the authors demonstrate that the length of the transcription bubble increases with the length between the -10 element and the TSS. Their determination of primer-concentration dependences of lifetimes of unwound states within open complexes yields the energy needed for scrunching of 1 or 2 bp and anti-scrunching of 1 bp. These values are consistent with the fact that for the vast majority of promoters, whose TSSs vary from the typical +1, these TSS occur at -1, +2 or +3. The authors argue that this energy consideration controls the TSS selection of all multi-subunit polymerases from bacterial, archaea, and eukaryotes and explains why the wide range of TSSs used by *S. cerevisiae* RNAP II requires ATP-hydrolysis of TFIIH. The current manuscript is an extension of their very recent publication in Science. However, the reported experiments are clear and the paper correlates the amount of scrunching/anti-scrunching with TSS by proper quantification. The paper will be of great interest to anyone who investigates fundamental mechanisms of transcription.

Comments:

1) The title is a little bit 'odd' because the reported phenomenon of scrunching/anti-scrunching do not select TSS. They accommodate the TSS selected by other factors. The author may select another title.

2) The authors do not discuss the factors of TSS selection published previously: The context sequence, the preference of a purine at+1, in the absence of a purine at +1 any presence of a purine nearby (-1, +2, +3). The reported discriminator triplet is not the major determinant.

3) It is so hard for the reader to know the TSS for the 64 discriminator sequences used here. I suggest the authors give a summary table for that.

4) The authors glossed over significant contribution of others relevant to TSS –

Winkelman et al., 2016 for example.

5) Is it okay to copy the Abstract more or less as the entire Introduction? I do not think so.

6) Typo subsection “TSS selection exhibits first hallmark of scrunching –movements of RNAP leading edge but not RNAP trailing edge – both in vitro and in vivo”: "…carrying genes for n engineered Bpa-specific…"

7) The logic surrounding the need for TFIIF by *S. cerevisiae* RNAP II is not clear since all polII's have TFIIF yet only *S. cerevisiae* scans widely for the TSS.

8) It would be useful for the authors to reference the Straney and Crothers paper (1987) in their background section as the original idea of a 'scrunched' (stressed) intermediate.

The manuscript is publishable somewhere. I do not recommend publication in its current form. I would like to see a revised manuscript addressing the comments I made above.

---

## [Author Response]

Reviewer #1:

In previous studies, Ebright and Nickels determined TSS and positions of leading and trailing edges of E. coli RNA polymerase from randomly mutagenized library of promotor sequences. They found a strong correlation between TSS and leading edge position, not between TSS and trailing edge, leading to the proposal of scrunching and antiscrunching for TSS selection downstream and upstream of canonical TSS, respectively. There are three main results reported here.1) They developed an in vivo crosslinking to demonstrate that the leading edge movement seen in vitro is also seen in vivo.2) They used magnetic tweezers to confirm that DNA unwinding is beyond the canonical amount for scrunching and vice versa. Importantly, they could show the change in DNA unwinding amount precisely corresponds to the distance by which TSS moves.3) Finally, they determined the lifetime of the triplet primer bound state to canonical, scrunched and anti-scrunched transcription bubble as a function of primer concentration, allowing them to estimate the energetic cost of scrunching and anti-scrunching. Satisfyingly, the energetic cost they determined can explain the observed distribution of TSS from the library they studied previously.Generally, the work is of high quality and provides additional validation to their prior proposals. There is no surprise or additional insight arising. The third component of this work provides a potentially important set of parameters, but what's unsatisfying is why certain sequences move TSS upstream vs downstream. Can the authors use the known sequence-dependent physical chemistry properties of DNA to predict which direction TSS moves?

We have added text at two places in subsection “TSS selection exhibits first hallmark of scrunching – movements of RNAP leading edge but not RNAP trailing edge – both in vitro and in vivo” summarizing published biochemical and structural results showing why different discriminator sequences move the TSS upstream or downstream:

"The results showed that the discriminator element (18,19) influences TSS selection…through effects on sequence-specific s-DNA interaction that select between two alternative paths of the DNA nontemplate strand (8)."

"For example, replacing a GGG discriminator by a CCT discriminator causes a 2 bp downstream change in TSS…from the position 7 bp downstream of the -10 element to the position 9 bp downstream of the -10 element, due to differences in sequence-specific σ-DNA interaction that result in different paths of the DNA nontemplate strand."

Reviewer #2:

[…] Subsection “TSS selection exhibits first hallmark of scrunching – movements of RNAP leading edge but not RNAP trailing edge – both in vitro and in vivo” from bottom carrying genes for *an* engineered.

We have corrected the typo.

Reviewer #3:[…] Comments:1) The title is a little bit 'odd' because the reported phenomenon of scrunching/anti-scrunching do not select TSS. They accommodate the TSS selected by other factors. The author may select another title.

We have changed the title to "The mechanism of variability in transcription start site selection."

2) The authors do not discuss the factors of TSS selection published previously: The context sequence, the preference of a purine at+1, in the absence of a purine at +1 any presence of a purine nearby (-1, +2, +3). The reported discriminator triplet is not the major determinant.

We have added text and literature citations in subsection “TSS selection exhibits first hallmark of scrunching – movements of RNAP leading edge but not RNAP trailing edge – both in vitro and in vivo” discussing the factors that determine TSS selection:

"TSS selection is a multi-factor process, in which the outcome reflects the contributions of promoter sequence and reaction conditions. TSS selection by bacterial RNAP and the bacterial transcription initiation factor σ involves four promoter-sequence determinants: (i) distance relative to the promoter -10 element (preference for TSS selection at the position 7 bp downstream of the promoter -10 element; 1-9); (ii) identities of the template-strand nucleotide at the TSS and the template-strand nucleotide immediately upstream of the TSS (strong preference for a template-strand pyrimidine at the TSS and preference for a template-strand purine immediately upstream of the TSS; 1-9); (iii) the promoter “core recognition element,” a segment of nontemplate-strand sequence spanning the TSS that interacts sequence-specifically with RNAP (preference for nontemplate-strand G immediately downstream of the TSS; 10), and (iv) the promoter “discriminator element,” a nontemplate-strand sequence immediately downstream of the promoter -10 element that interacts sequence-specifically with σ (preference for TSS selection at upstream positions for purine-rich discriminator sequences, and preference for TSS selection at downstream positions for pyrimidine-rich discriminator sequences) (8,9) In addition to these four promoter-sequence determinants, the concentrations of initiating NTPs (2,4,5,7,11-16) and DNA topology (7) also influence TSS selection."

3) It is so hard for the reader to know the TSS for the 64 discriminator sequences used here. I suggest the authors give a summary table for that.

Although a previous paper from our group discussed 64 discriminator sequences, this manuscript discusses only two discriminator sequences: GGG and CCT. The properties of these two discriminator sequences are summarized in subsection “TSS selection exhibits first hallmark of scrunching – movements of RNAP leading edge but not RNAP trailing edge – both in vitro and in vivo”:

"For example, replacing a GGG discriminator by a CCT discriminator causes a 2 bp downstream change in TSS (from the position 7 bp downstream of the -10 element to the position 9 bp downstream of the -10 element, due to differences in sequence-specific σ-DNA interaction that result in different paths of the DNA nontemplate strand), causes a 2 bp downstream change in RNAP leading-edge position, but does not cause a change in RNAP trailing-edge position (Figure 1).”

4) The authors glossed over significant contribution of others relevant to TSS –Winkelman et al., 2016 for example.

We have added text and literature citations in subsection “TSS selection exhibits first hallmark of scrunching – movements of RNAP leading edge but not RNAP trailing edge – both in vitro and in vivo” discussing the factors that determine TSS selection:

"TSS selection is a multi-factor process, in which the outcome reflects the contributions of promoter sequence and reaction conditions. TSS selection by bacterial RNAP and the bacterial transcription initiation factor σ involves four promoter-sequence determinants: (i) distance relative to the promoter -10 element (preference for TSS selection at the position 7 bp downstream of the promoter -10 element; 1-9); (ii) identities of the template-strand nucleotide at the TSS and the template-strand nucleotide immediately upstream of the TSS (strong preference for a template-strand pyrimidine at the TSS and preference for a template-strand purine immediately upstream of the TSS; 1-9); (iii) the promoter “core recognition element,” a segment of nontemplate-strand sequence spanning the TSS that interacts sequence-specifically with RNAP (preference for nontemplate-strand G immediately downstream of the TSS; 10), and (iv) the promoter “discriminator element,” a nontemplate-strand sequence immediately downstream of the promoter -10 element that interacts sequence-specifically with σ (preference for TSS selection at upstream positions for purine-rich discriminator sequences, and preference for TSS selection at downstream positions for pyrimidine-rich discriminator sequences) (8,9). In addition to these four promoter-sequence determinants, the concentrations of initiating NTPs (2,4,5,7,11-16) and DNA topology (7) also influence TSS selection."

5) Is it okay to copy the Abstract more or less as the entire Introduction? I do not think so.

We have significantly expanded the Introduction, adding (i) text and literature citations discussing the factors that determine TSS selection, and (ii) text and literature citations on the hypothesis tested, and previous results supporting the hypothesis, and new results supporting the hypothesis:

"TSS selection is a multi-factor process, in which the outcome reflects the contributions of promoter sequence and reaction conditions. TSS selection by bacterial RNAP and the bacterial transcription initiation factor σ involves four promoter-sequence determinants: (i) distance relative to the promoter -10 element (preference for TSS selection at the position 7 bp downstream of the promoter -10 element; 1-9); (ii) identities of the template-strand nucleotide at the TSS and the template-strand nucleotide immediately upstream of the TSS (strong preference for a template-strand pyrimidine at the TSS and preference for a template-strand purine immediately upstream of the TSS; 1-9); (iii) the promoter “core recognition element,” a segment of nontemplate-strand sequence spanning the TSS that interacts sequence-specifically with RNAP (preference for nontemplate-strand G immediately downstream of the TSS; 10), and (iv) the promoter “discriminator element,” a nontemplate-strand sequence immediately downstream of the promoter -10 element that interacts sequence-specifically with σ (preference for TSS selection at upstream positions for purine-rich discriminator sequences, and preference for TSS selection at downstream positions for pyrimidine-rich discriminator sequences) (8,9) In addition to these four promoter-sequence determinants, the concentrations of initiating NTPs (2,4,5,7,11-16) and DNA topology (7) also influence TSS selection."

"It has been hypothesized that variability in the distance between core promoter elements and the TSS is accommodated by DNA "scrunching" and "anti-scrunching," the defining hallmarks of which are: (i) forward and reverse movements of the RNAP leading edge, but not the RNAP trailing edge, relative to DNA and (ii) expansion and contraction of the transcription bubble (7-10, 17).In previous work, we showed that TSS selection exhibits the first hallmark of scrunching in vitro (8). Here, we show that TSS selection also exhibits the first hallmark of scrunching in vivo, show that TSS selection exhibits the second hallmark of scrunching and anti-scrunching, and define the energetics of scrunching and anti-scrunching."

6) Typo subsection “TSS selection exhibits first hallmark of scrunching--movements of RNAP leading edge but not RNAP trailing edge--both in vitro and in vivo”: "…carrying genes for n engineered Bpa-specific…"

We have corrected the typo.

7) The logic surrounding the need for TFIIF by S. cerevisiae RNAP II is not clear since all polII's have TFIIF yet only S. cerevisiae scans widely for the TSS.

The manuscript suggests a role for TFIIH, not for TFIIF, in TSS scanning by *S. cerevisiae* RNAP II.

With respect to the basis for species differences in TSS scanning by RNAP II, published work indicates that species differences in TSS scanning are attributable to species differences in RNAP II and TFIIB (Li et al., 1994; https://www.ncbi.nlm.nih.gov/pubmed/?term=8303296).

8) It would be useful for the authors to reference the Straney and Crothers paper (1987) in their background section as the original idea of a 'scrunched' (stressed) intermediate.

We have added text and literature citations (including Straney and Crothers, 1979) in subsection “Energetic costs of scrunching and anti-scrunching” discussing implications of the results for the energetics of the "stressed intermediate" in initial transcription:

"We hypothesize that energetic costs on the same scale, ~0.7-1.8 kcal/mol per scrunched bp, also apply in the structurally and mechanistically related scrunching that occurs during initial transcription by RNAP (24,26). We note that, according to this hypothesis, the scrunching by ~10 bp that occurs during initial transcription (24) results in an increase in the state energy of the transcription initiation complex by a total of ~7-18 kcal/mol (~10 x ~0.7-1.8 kcal/mol). This is an increase in state energy potentially sufficient to yield a "stressed intermediate" (24,27) having scrunching-dependent "stress" comparable to the free energies of RNAP-promoter and RNAP-initiation-factor interactions that anchor RNAP at a promoter (~7-9 kcal/mol for sequence-specific component of RNAP-promoter interaction and ~13 kcal/mol for RNAP-initiation-factor interaction; 24) and therefore is an increase in state energy potentially sufficient to pay energetic costs of breaking RNAP-promoter and RNAP-initiation-factor interactions in the transition from transcription initiation to transcription elongation.”